# From Inpainting to Editing:
# Unlocking Robust Mask-Free Visual Dubbing via Generative Bootstrapping

**Xu He** [1] **Haoxian Zhang** [2] **Hejia Chen** [2] **Changyuan Zheng** [1] **Liyang Chen** [1] **Songlin Tang** [2] **Jiehui Huang** [3]
**Xiaoqiang Liu** [2] **Pengfei Wan** [2] **Zhiyong Wu** [1 4]

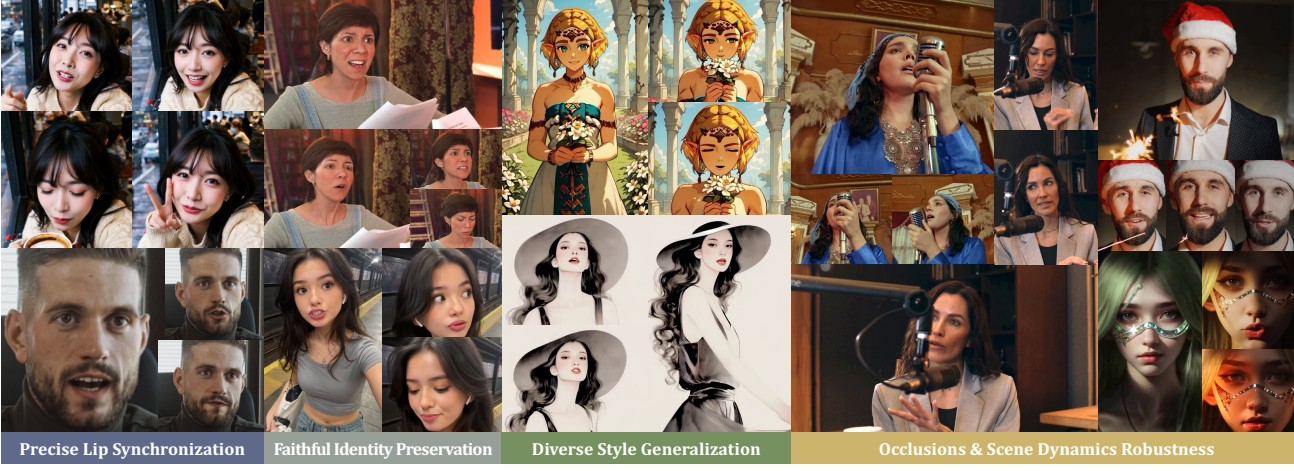

*Figure 1.* Our proposed generative bootstrapping framework unlocks high-fidelity mask-free visual dubbing, delivering precise lip sync and faithful identity preservation, even in challenging scenarios with occlusions and dynamic lighting.

## Abstract

Audio-driven visual dubbing aims to synchronize a video's lip movements with new speech but is fundamentally challenged by the lack of ideal training data: paired videos differing only in lip motion. Existing methods circumvent this via mask-based inpainting. However, masking inevitably destroys spatiotemporal context, leading to identity drift and poor robustness (e.g., to occlusions), while also inducing lip-shape leakage that degrades lip sync. To bridge this gap, we propose **X-Dub**, a novel two-stage generative bootstrapping framework leveraging powerful Diffusion Transformers to unlock mask-free dubbing. Our core insight is to repurpose a mask-based inpainting model exclusively as a dedicated data generator to synthesize scalable, high-fidelity pseudo-paired data, which is subsequently utilized to train and bootstrap a robust, mask-free editing model as the final video dubber. The final dubber is liberated from masking artifacts and leverages the complete video input for high-fidelity inference. We further introduce timestep-adaptive multi-phase learning to disentangle conflicting objectives (structure, lip motion, and texture) across diffusion phases, facilitating stable convergence and advanced editing quality. Additionally, we present X-DubBench, a benchmark for diverse scenarios. Extensive experiments demonstrate that our method achieves state-of-the-art performance with superior lip sync, visual quality, and robustness. Code, demos, and additional resources are available at https://github.com/KlingAIResearch/X-Dub.

[1]Tsinghua Shenzhen International Graduate School, Tsinghua University, Shenzhen, China [2]Kling Team, Kuaishou Technology, Shenzhen, China [3]The Hong Kong University of Science and Technology, Hong Kong SAR, China [4]The Chinese University of Hong Kong, Hong Kong SAR, China. Correspondence to: Zhiyong Wu <zywu@sz.tsinghua.edu.cn>.

*Proceedings of the 43rd International Conference on Machine Learning*, Seoul, South Korea. PMLR 306, 2026. Copyright 2026 by the author(s).

## 1. Introduction

Audio-driven visual dubbing aims to synchronize an existing video's lip movements with new speech (Prajwal et al., 2019), demonstrating broad applications from personalized avatars (Thies et al., 2020) to multilingual film

translation (Prajwal et al., 2020). While recent advances in Diffusion Transformers (DiTs) (Peebles & Xie, 2023) have accelerated progress in audio-driven portrait animation (Chen et al., 2025b; Lin et al., 2025a;b; Cui et al., 2026) by enabling high-fidelity video synthesis (Zhou et al., 2026; Cai et al., 2025b;a), these methods predominantly focus on generating entire videos from a single still image. Visual dubbing, however, poses unique challenges: it requires precise modification of speech-relevant facial regions while faithfully preserving all other visual attributes (e.g., identity, head poses, and scene dynamics), thereby ensuring seamless integration into the source footage (Guan et al., 2023).

Existing approaches predominantly rely on *mask-guided inpainting* (Prajwal et al., 2020; Li et al., 2024), where the mouth region in source frames is masked and subsequently inpainted conditioned on speech. While this design conveniently enables self-supervised training on unlabeled videos, it introduces intrinsic limitations. First, manually designed masks inevitably induce lip-shape leakage, either through movements of adjacent regions (e.g., cheeks, jaws) or via adaptive mask boundaries (Pan et al., 2025). This leakage is further exacerbated by audio's inferior conditioning capacity relative to visual cues, driving the model towards visual-to-visual shortcut learning (Bigata et al., 2025) that fundamentally compromises lip sync during inference. Moreover, masking physically destroys the spatiotemporal context of the provided video, stripping away essential identity and environmental cues. Although these methods attempt to compensate using reference frames from other segments, the model is still forced to hallucinate missing content (e.g., facial occlusions, shifting lighting, and shadows) and extract identity from pose-misaligned references. This inherently leads to identity drift and visual artifacts, particularly when target poses or scene dynamics diverge significantly from the references (Zhong et al., 2023; Peng et al., 2025).

Ideally, these limitations could be overcome by formulating visual dubbing as a direct audio-driven *video-to-video (V2V) editing* task, bypassing explicit masks and their constraints. However, training such a model necessitates paired video data: two videos identical in identity, pose, and environment, differing only in lip motion. In reality, obtaining such pairs from real footage is virtually impossible. While synthetic alternatives like 3D renderings (Chen et al., 2025a) exist, they often lack the photorealistic diversity and scalability required to generalize to complex real-world scenarios. This data scarcity remains the primary bottleneck preventing the adoption of robust V2V frameworks in visual dubbing.

To bridge this gap, we propose X-Dub, a novel **generative bootstrapping framework** built upon a powerful pretrained DiT backbone, that finally unlocks mask-free visual dubbing through a two-stage learning paradigm. Our core insight is to relegate a mask-guided inpainting-based model

to the role of a dedicated *data generator*, utilizing it to synthesize scalable paired pseudo-training data (**Stage I**), which is subsequently utilized to train and bootstrap a robust, mask-free editing-based final *video dubber* for final inference (**Stage II**). Consequently, the final video dubber is liberated from the adverse effects of masking, such as lip-shape leakage, and can perform inference by directly leveraging the complete input video as comprehensive spatiotemporal context. This design offers a dual benefit. On the one hand, by harnessing the generative capability of DiT, we produce highly photorealistic and scalable training pairs directly from real-world footage, overcoming the scalability limits and domain gaps of alternative synthetic data sources (e.g., 3D renderings). On the other hand, instead of enforcing mask constraints during unpredictable final inference as other mask-based methods do, we strategically shift the burden of masking and its associated artifacts to a controllable data construction phase. This allows potential leakage or instability to be mitigated and filtered out during data preparation, rather than manifesting in the final inference output, ultimately yielding superior mask-free dubbing results.

Specifically, in the **first stage (Data Construction)**, we develop a mask-guided inpainting model trained on large-scale unlabeled audiovisual data, functioning exclusively as a *data generator*. Once trained, it replaces the audio of real source videos with alternative speech to generate corresponding dubbed counterparts, forming **synthetic-real video data pairs** where only lip movements differ. Acknowledging that mask-based generation is not always perfect, we tailor specific mitigation measures to this controllable, in-domain data preparation phase. Dedicated data creation, filtering, and augmentation strategies are designed to foster a curated dataset that is high-quality, diverse, and closely aligned with real-world distributions, thereby laying a robust foundation for the subsequent mask-free training. In the **second stage (Mask-Free Dubbing)**, the mask-free editing-based *video dubber* is trained to predict the authentic source video from its synthetic counterpart, conditioned on the original speech. By consistently utilizing real videos as the supervision target, we effectively anchor the editor's output distribution to real-world data, preventing the learning of synthesis errors while alleviating the strict requirement for precise lip-sync in the generated training input. Moreover, training on these potentially imperfect synthetic inputs implicitly enhances the model's robustness against artifacts in complex, in-the-wild scenarios. It is worth noting that our intention lies not merely in an advanced network architecture, but in exploring a novel **learning paradigm** for high-quality visual dubbing by constructing pseudo-paired videos from abundant, unlabeled, in-the-wild audiovisual resources.

However, effectively training this mask-free editor in the second stage presents a distinct optimization challenge: the model must simultaneously perform lip editing while main-

taining rigorous global structure (e.g., head pose and background layout) and fine-grained texture preservation (e.g., skin details), which span vastly different spatiotemporal frequencies. Monolithic training often struggles to balance these competing objectives, leading to unstable convergence in our early experiments. To address this, we introduce a **timestep-adaptive multi-phase learning strategy**. Leveraging the inherent frequency bias of diffusion models which capture distinct information levels at different denoising timesteps (Zhang et al., 2025; Wang et al., 2025), we decouple the training process into progressive phases, aligning high, mid, and low noise levels with structure, lip motion, and texture refinement, respectively. This "divide-and-conquer" design facilitates training and also enables targeted supervision (e.g., lip-sync and identity reward) at optimal intervals, enhancing editing quality while to some extent compensating for potential synthetic data imperfections.

Finally, to rigorously assess visual dubbing performance in more complex scenarios, we introduce X-DubBench, a benchmark comprising diverse real-world footage and high-quality AI-generated videos, covering varied motions and environments beyond the scope of existing lab-controlled datasets (Afouras et al., 2018; Zhang et al., 2021).

To summarize, our main contributions are: 1) We propose a **generative bootstrapping framework** for visual dubbing. By repurposing a mask-guided inpainting model to synthesize pseudo-paired data, we bootstrap a robust, mask-free editing model as the final video dubber, fundamentally eliminating mask artifacts and achieving superior lip sync and robustness across diverse complex scenarios. 2) We introduce a **timestep-adaptive multi-phase learning strategy** to disentangle conflicting editing objectives across diffusion timesteps, facilitating training convergence and enhancing lip sync and visual fidelity. 3) We release **X-DubBench**, a diverse benchmark specifically designed for evaluating dubbing in challenging practical scenarios. 4) Extensive experiments demonstrate that our method achieves state-of-the-art performance, significantly outperforming existing approaches across comprehensive metrics regarding lip sync accuracy, visual fidelity, and identity preservation.

## 2. Related Work

**Visual Dubbing.** Early visual dubbing methods leverage GANs (Goodfellow et al., 2014) for mask-based inpainting. LipGAN (Prajwal et al., 2019) pioneers reference-guided synthesis, while Wav2Lip (Prajwal et al., 2020) improves lip sync via SyncNet (Chung & Zisserman, 2016). Subsequent works extend this paradigm by addressing expression bias, resolution, and intelligibility, including VideoReTalking (Cheng et al., 2022), DINet (Zhang et al., 2023), and TalkLip (Wang et al., 2023), with IP-LAP (Zhong et al., 2023) and StyleSync (Guan et al., 2023) improving identity

preservation. Recent diffusion-based approaches demonstrate stronger generation capability. DiffTalk (Shen et al., 2023) and Diff2Lip (Mukhopadhyay et al., 2024) validate diffusion for visual dubbing, while MuseTalk (Zhang et al., 2024) and LatentSync (Li et al., 2024) improve efficiency and temporal stability. Nevertheless, existing methods remain bound to a *mask-guided* inpainting workflow, where explicit masks often cause lip-sync errors and visual artifacts, particularly under occlusions or large head motions. By contrast, we restrict mask-based inpainting to a controllable data construction stage for synthesizing pseudo-paired data, which bootstraps a superior *mask-free* video dubber and eliminates mask-induced constraints during inference.

**Audio-Driven Portrait Animation.** Another related line of work is audio-driven portrait animation, which generates talking videos from still images or text prompts. Recent DiT-based models achieve expressive talking-head (Tian et al., 2024; Cui et al., 2024a), half-body (Cui et al., 2024b; Meng et al., 2025), and full-body results (Wang et al., 2025; Lin et al., 2025a). These works demonstrate the power of DiTs for human-centric video generation. Visual dubbing instead is a stricter V2V editing task: it requires precise speech-driven modifications while preserving other visual cues, enabling seamless integration into recorded videos.

## 3. Our Approach

Fig. 2 illustrates our generative bootstrapping framework. Unlike prevalent methods that directly deploy mask-guided models for dubbing inference, which often yield degraded lip sync and visual artifacts, we adopt a two-stage strategy: a mask-guided model functions solely as a *data generator* to construct realistic pseudo-paired training data within a controllable scope, which then bootstraps a superior, mask-free *video dubber* for in-the-wild inference. In Sec. 3.1 (**Stage I**), we first introduce the audio-driven *mask-based inpainting model*, featuring tailored designs to function as a specialized *data generator*. We then detail the synthesis of highly realistic pseudo-paired data through a dedicated pipeline comprising creation, filtering, and augmentation strategies, laying a robust foundation for the subsequent training. In Sec. 3.2 (**Stage II**), we present how we utilize this curated dataset to bootstrap a *mask-free editing model*, which functions as the robust *video dubber* during final inference. Finally, in Sec. 3.3, we detail a timestep-adaptive multi-phase learning scheme, which facilitates stable training and further enhances the editor's capabilities by disentangling conflicting objectives across diffusion timesteps.

**DiT Backbone.** Our DiT backbone follows the latent diffusion paradigm with a 3D VAE for video compression and a DiT for sequence modeling (Peebles & Xie, 2023). Each DiT block combines 2D spatial and 3D spatio-temporal self-attention with cross-attention for external conditions.

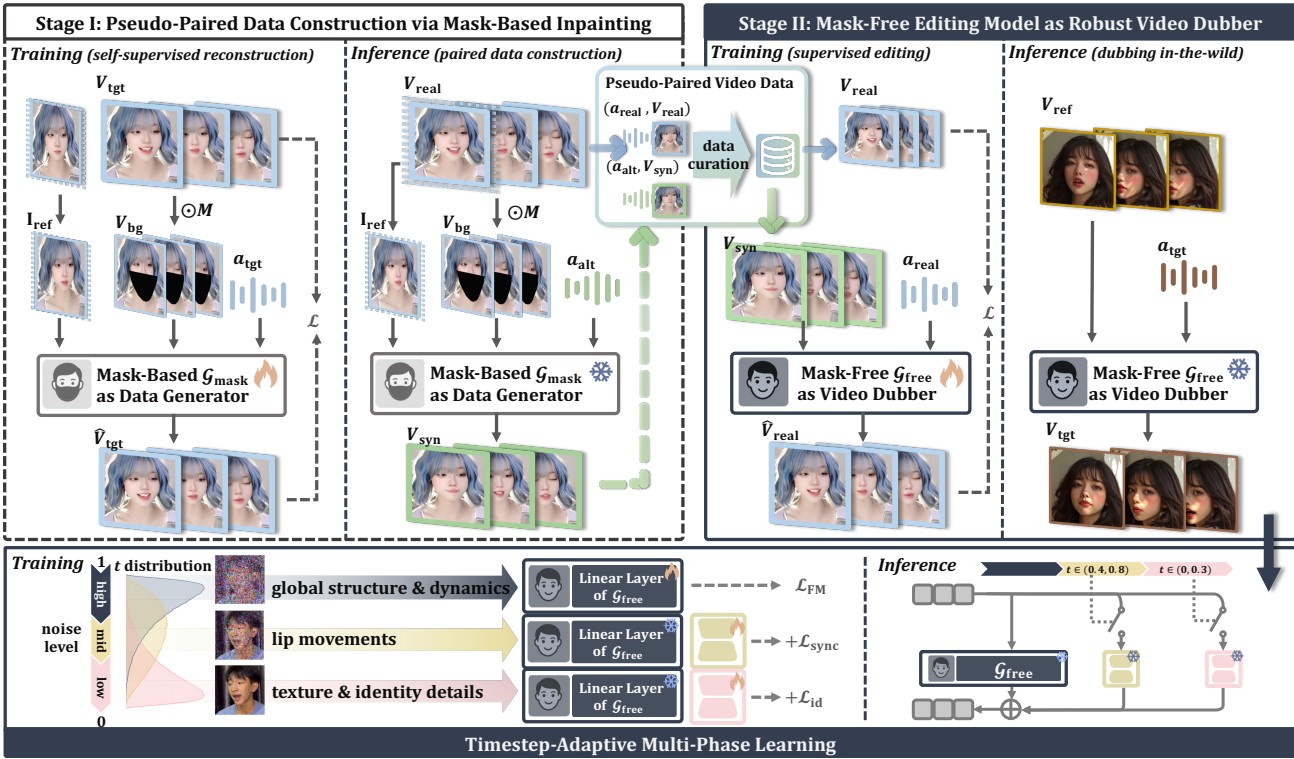

*Figure 2.* **Overview of X-Dub. Stage I (Data Construction):** We formulate a mask-guided inpainter as a data generator to create pseudo-paired data (identical context, altered lips) via a dedicated curation pipeline. **Stage II (Mask-Free Dubbing):** These pairs bootstrap a mask-free video dubber that learns to dub directly from complete video inputs, overcoming mask limitations. Crucially, our **multi-phase strategy** disentangles training by aligning specific diffusion phases with structure, lip, and texture objectives.

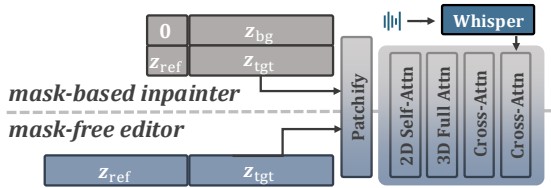

*Figure 3.* **Conditioning structure.** References (sparse frames for inpainter; full video for editor) are concatenated with the target for 3D self-attention, while audio is injected via cross-attention.

## 3.1. Stage I: Pseudo-Paired Data Construction with Mask-Based Data Generator

In this stage, we first establish an audio-driven mask-based inpainting model, denoted as $\mathcal{G}_{\text{mask}}$, and train it on extensive unlabeled audiovisual data in a self-supervised manner. Once trained, $\mathcal{G}_{\text{mask}}$ functions as a specialized data generator to synthesize highly realistic pseudo-paired data through a tailored in-domain data preparation pipeline, laying the foundation for the subsequent mask-free stage.

**Audio-Driven Mask-Guided Inpainting Model.** $\mathcal{G}_{\text{mask}}$ is trained under a self-supervised reconstruction paradigm. Given a target video clip $V_{\text{tgt}}$ and its corresponding audio $a_{\text{tgt}}$, we mask the lower facial regions using a binary mask

$M$ to obtain the background frames $V_{\text{bg}} = V_{\text{tgt}} \odot M$. The model takes $V_{\text{bg}}$, $a_{\text{tgt}}$, and $N$ reference frames $\{I_{\text{ref}}^i\}_{i=1}^N$ randomly sampled from different segments of the same video as input to reconstruct the target video, yielding $\hat{V}_{\text{tgt}}$.

Specifically, as illustrated in Fig. 3, the background frames $V_{\text{bg}}$, target frames $V_{\text{tgt}}$, and reference frames are encoded by the VAE into latents $z_{\text{bg}}, z_{\text{tgt}} \in \mathbb{R}^{b \times f \times c \times h \times w}$ and $z_{\text{ref}} \in \mathbb{R}^{b \times N \times c \times h \times w}$, respectively. We concatenate $z_{\text{bg}}$ channel-wise with the noised target latent $z_{\text{tgt}}$ (or pure noise during inference). To align dimensions, we channel-pad the reference latents with zeros. The unified input sequence $z_{\text{in}}$ is constructed by concatenating these components along the frame dimension: $z_{\text{in}} = \left[ [0, z_{\text{ref}}]_{\text{ch}}, [z_{\text{bg}}, z_{\text{tgt}}]_{\text{ch}} \right]_{\text{fr}}$, enabling the DiT to model interactions between the target context and reference identity via 3D self-attention. For audio conditioning, we extract audio embeddings from $a_{\text{tgt}}$ using a pre-trained Whisper (Radford et al., 2023) encoder and inject them into the DiT blocks via cross-attention.

The training of $\mathcal{G}_{\text{mask}}$ is governed by a flow-matching loss $\mathcal{L}_{\text{FM}}$, spatially weighted by face masks $M_{\text{face}}$ and lip masks $M_{\text{lip}}$ derived from DWPose (Yang et al., 2023):

$$\mathcal{L}_{\text{wFM}} = (1 + \lambda_{\text{face}} M_{\text{face}} + \lambda_{\text{lip}} M_{\text{lip}}) \odot \mathcal{L}_{\text{FM}}. \quad (1)$$

To support long-video generation, we employ motion-frame-based concatenation (Tian et al., 2024): the generation of each segment is conditioned on the last $m = 2$ frames of the preceding segment. During training, the first $m$ frames of $z_{\text{tgt}}$ remain unnoised to serve as motion guidance.

**Tailored Design as a Data Generator.** Functioning as a specialized data generator in our framework rather than a generic dubbing model, $\mathcal{G}_{\text{mask}}$ is tailored to faithfully preserve visual attributes like identity and color from the source video. Conversely, lip sync accuracy is less emphasized, as the output of $\mathcal{G}_{\text{mask}}$ never serves as supervision in Stage II, and any lip imperfections will not be propagated to our final dubber. In our initial empirical observations, we found that relying on a single reference frame (Li et al., 2024) or generating long clips directly in a single pass with the mask-based $\mathcal{G}_{\text{mask}}$ often results in non-negligible identity and color drift. Therefore, we condition $\mathcal{G}_{\text{mask}}$ on multiple reference frames and restrict it to generate shorter segments than the final video dubber. These segments are then concatenated via motion frames to form complete synthetic video data with minimized visual drift, albeit at the potential cost of degraded lip sync. Further details are provided in Appendix B.

**Dedicated Data Construction Pipeline.** Leveraging the well-trained and tailored $\mathcal{G}_{\text{mask}}$, we establish a robust pipeline to construct pseudo-paired data. For each real video clip $V_{\text{real}}$, we replace its audio $a_{\text{real}}$ with an alternative track $a_{\text{alt}}$ to synthesize a counterpart $V_{\text{syn}}$. Specifically, we restore the original background in $V_{\text{syn}}$ with boundary fusion, eliminating potential background artifacts. This process yields aligned pair videos $(V_{\text{syn}}, V_{\text{real}})$ identical in identity, pose, and background but differing in lip motion. To promote the quality and diversity, we design dedicated strategies encompassing: **1) In-domain creation.** We perform data synthesis strictly within the training domain of $\mathcal{G}_{\text{mask}}$ and sample $a_{\text{alt}}$ from the same speaker as $V_{\text{real}}$, avoiding instability from unseen data or cross-identity audio-visual combinations. **2) Quality filtering.** We perform multi-dimensional quality filtering using landmark distance, identity similarity, and visual quality scores to ensure sufficient lip divergence, identity preservation, and high visual fidelity. **3) Augmentation.** To prepare the mask-free editor for challenging scenarios, we augment the curated pairs with diverse occlusions and lighting conditions, compensating for complex samples that may have been excluded during the filtering process. Furthermore, we additionally supplement the dataset with a subset of 3D-rendered data featuring perfectly-aligned identity, scene, and pose to anchor precise lip editing. Data construction details can be found in Appendix B. With these designs together, we establish a highly realistic and scalable pseudo-paired video dataset derived from widely available unlabeled real-world data.

## 3.2. Stage II: Robust Mask-Free Video Dubber

Leveraging the curated pseudo-paired dataset from Stage I, we train and bootstrap a mask-free editing model $\mathcal{G}_{\text{free}}$, which is the ultimate goal and final executor of our video dubbing framework. Driven by aligned pairs, $\mathcal{G}_{\text{free}}$ autonomously learns to locate and edit speech-relevant facial regions, thereby fundamentally eliminating boundary leakage and artifacts inherent to mask-based approaches.

Structurally, as shown in Fig. 3, we encode paired reference and target videos into latents $z_{\text{ref}}, z_{\text{tgt}}$. The clean $z_{\text{ref}}$ is concatenated with the noised $z_{\text{tgt}}$ (or pure noise during inference) along the frame axis to form the input $z_{\text{in}} \in \mathbb{R}^{b \times 2f \times c \times h \times w}$. Patchifying this combined sequence enables contextual interaction via 3D self-attention, which minimally alters the DiT backbone yet fully exploits its contextual modeling capacity. Audio features and motion frames are integrated identically to the protocol in Sec. 3.1.

During training, we consistently use the real video $V_{\text{real}}$ from the curated pairs as the supervision target and its synthesized counterpart $V_{\text{syn}}$ as the reference input. This strategy prevents artifacts remaining in the curated data from propagating to the video dubber $\mathcal{G}_{\text{free}}$, and also relieves the data generator $\mathcal{G}_{\text{mask}}$ of the burden of perfect lip accuracy in stage I. At inference, $\mathcal{G}_{\text{free}}$ directly processes user-provided real videos, leveraging full spatiotemporal contexts without mask-induced degradation. Moreover, training on potentially imperfect synthetic inputs with high-quality real supervision to some extent enhances the model's robustness against noise and artifacts in complex, in-the-wild scenarios.

## 3.3. Timestep-Adaptive Multi-Phase Learning

While the pseudo-paired data raises the performance ceiling of visual dubbing, training a robust mask-free editing model poses unique challenges. Specifically, it requires the model to autonomously learn and balance potentially conflicting objectives: inheriting global spatiotemporal structure, precisely editing lip motion, and preserving fine-grained identity details. Observing that diffusion models exhibit phase-wise specialization across timesteps (Zhang et al., 2025; Wang et al., 2025), we are motivated to introduce a timestep-adaptive multi-phase scheme, where different noise regions target these complementary objectives.

**Training Phase Partitioning.** During training, instead of sampling timesteps uniformly, we follow Esser et al. (2024) to shift the sampling distribution to concentrate on different noise levels for distinct training phases:

$$t_{\text{shift}} = \alpha t_{\text{base}} / (1 + (\alpha - 1) t_{\text{base}}), \tag{2}$$

where $t_{\text{base}}$ is logit-normal and $\alpha$ sets the shift strength. This allows us to target: 1) high-noise steps for global structure and motion (e.g., background, pose, overall contours); 2)

mid-noise steps for lip movements; 3) low-noise steps for texture refinement concerning identity details.

**High-Noise Full Training.** We first optimize the editor under a high-noise distribution via full-parameter tuning. This fosters convergence (Esser et al., 2024) and structural learning, seamlessly transferring global dynamics (e.g., background, pose) from the input while achieving preliminary lip sync. The objective remains $\mathcal{L}_{\text{wFM}}$ (Eq. 1).

**Mid-Noise and Low-Noise Tuning with LoRA Experts.** We then attach lightweight LoRA modules for mid- and low-noise tuning. To enable pixel-level supervision without training overhead, we design a single-step denoising strategy:

$$\hat{x}_0 = \mathcal{D}(z_0 + (v - \hat{v}) \cdot \tau), \tag{3}$$

where $\tau = t$ if $t \leq t_{\text{thres}}$, and $\tau = t_{\text{thres}}$ otherwise. This truncation ensures stable denoising at high noise levels (see Appendix C for detailed derivation).

The *lip expert* operates at mid-noise, where we incorporate an auxiliary SyncNet loss $\mathcal{L}_{\text{sync}}$ to enforce audio-visual alignment. The *texture expert* functions at low-noise, additionally supervised by identity loss $\mathcal{L}_{\text{id}}$ (Deng et al., 2019; Radford et al., 2021) computed against references to refine fidelity. To avoid hurting sync, we randomly disable audio cross-attention ($p = 0.5$) during texture tuning, applying texture supervision only on the silent branches.

During inference, we activate the texture ($t \in [0, 0.3]$) and lip ($t \in [0.4, 0.8]$) experts within their most effective ranges. Timestep selection details are provided in Appendix C.3.

# 4. Experiments

**Benchmark.** To evaluate visual dubbing in practical settings, we construct X-DubBench, a challenging benchmark of 440 video-audio pairs combining real-world and AI-generated content. Videos include challenging scenarios like pose changes, occlusions, and stylizations, while audio covers speech and singing across six languages. Unlike existing controlled datasets, it enables evaluation under complex, realistic conditions, as detailed in Appendix J.

**Evaluation metrics.** We evaluate generation quality using PSNR, SSIM, Fréchet Inception Distance (FID) for spatial quality, and Fréchet Video Distance (FVD) for temporal consistency. Lip-sync quality is measured by landmark distance (LMD) and SyncNet confidence (Sync-C). Identity preservation is assessed through cosine similarity of ArcFace embeddings (CSIM), CLIP score (CLIPS) for semantic features, and LPIPS for perceptual similarity. For the more challenging X-DubBench, we additionally report no-reference perceptual metrics, including Natural Image Quality Evaluator (NIQE), Blind/Referenceless Image Spatial Quality Evaluator (BRISQUE), and HyperIQA (Su et al.,

2020). We also report the success rate over all video samples, which is crucial in practice, as many mask-based methods completely fail under challenging scenarios.

## 4.1. Quantitative Evaluation

We evaluate our mask-free video dubber $\mathcal{G}_{\text{free}}$ on both HDTF (Zhang et al., 2021) and X-DubBench, comparing against state-of-the-art methods including Wav2Lip (Prajwal et al., 2020), VideoReTalking (Cheng et al., 2022), TalkLip (Wang et al., 2023), IP-LAP (Zhong et al., 2023), Diff2Lip (Mukhopadhyay et al., 2024), MuseTalk (Zhang et al., 2024), and LatentSync (Li et al., 2024). Baseline methods are evaluated using their official pretrained weights, consistent with common practice in prior work. We additionally retrain LatentSync, the strongest overall baseline, on the same 600-hour internet video corpus used for our data construction (denoted as LatentSync†). To isolate paradigm gains from backbone capacity, we additionally reimplement a generic variant of our mask-based data generator, noted as $\mathcal{G}_{\text{mask}}^*$, by removing its data-creation constraints. This allows for a fair comparison between mask-guided inpainting and our mask-free editing approach.

Tables 1 and 2 show that $\mathcal{G}_{\text{free}}$ establishes a new state-of-the-art. On HDTF, it achieves significant gains in visual quality (FID –12.6%), lip sync (Sync-C +4.9%), and identity retention (CSIM +4.3%). These gains are amplified on the challenging X-DubBench, where $\mathcal{G}_{\text{free}}$ delivers superior visual scores (NIQE 5.78, BRISQUE 29.9), lip sync (Sync-C +16.0%), and identity preservation (CSIM +6.1%). Notably, it attains a 96.4% success rate (+24 points over the strongest baseline), highlighting the robustness of our mask-free approach in unconstrained scenarios.

Crucially, while our mask-based $\mathcal{G}_{\text{mask}}^*$ already surpasses priors on HDTF (CLIPS +1.7%, FVD –26.8%), confirming the backbone's capacity for synthesizing realistic pseudo-pairs, our mask-free $\mathcal{G}_{\text{free}}$ achieves further improvements (CSIM +3.3%, Sync-C +6.4%, LPIPS –22.2%) while maintaining comparable FVD. This effectively validates our paradigm, demonstrating that data synthesized by the mask-based generator successfully bootstraps a superior mask-free dubber.

## 4.2. Qualitative Evaluation

Fig. 4 presents qualitative comparisons, where our method consistently produces realistic, lip-synced results under challenging scenarios. Mask-based baselines frequently suffer from inaccurate lip shapes (Col. 1), visual artifacts (Col. 2), poor occlusion robustness (Col. 5), and side-view distortions with identity drift (Col. 2&9). Even our $\mathcal{G}_{\text{mask}}^*$, despite its powerful DiT backbone, exhibits blurring around occlusions. Notably, the rightmost column reveals severe lip-shape leakage in all mask-based methods, corrupting silent frames with open-mouth artifacts. In contrast, our

*Table 1.* **Quantitative results on the HDTF dataset.** Top three results are highlighted as first , second , and third . LatentSync†
denotes LatentSync retrained on the same 600-hour internet video corpus.

| | HDTF Dataset | | | | | | | | |
|---|---|---|---|---|---|---|---|---|---|
| | Visual Quality | | | | Lip Sync | | Identity | | |
| Method | PSNR ↑ | SSIM ↑ | FID ↓ | FVD ↓ | Sync-C ↑ | LMD ↓ | LPIPS ↓ | CSIM ↑ | CLIPS ↑ |
| Wav2Lip | 27.412 | 0.851 | 15.475 | 530.905 | 7.663 | 0.896 | 0.078 | 0.807 | 0.842 |
| VideoReTalking | 25.189 | 0.844 | 11.303 | 327.886 | 7.482 | 1.170 | 0.056 | 0.745 | 0.808 |
| TalkLip | 27.024 | 0.850 | 17.315 | 564.307 | 5.887 | 0.858 | 0.060 | 0.804 | 0.855 |
| IP-LAP | 28.571 | 0.860 | 9.026 | 352.403 | 5.199 | 0.934 | 0.041 | 0.840 | 0.899 |
| Diff2Lip | 28.716 | 0.860 | 12.251 | 348.290 | 7.897 | 0.911 | 0.036 | 0.790 | 0.876 |
| MuseTalk | 29.542 | 0.866 | 8.123 | 258.236 | 6.409 | 0.741 | 0.029 | 0.824 | 0.884 |
| LatentSync | 31.325 | 0.903 | 8.042 | 235.524 | 8.163 | 0.821 | 0.024 | 0.847 | 0.902 |
| LatentSync† | 31.718 | 0.908 | 7.973 | 214.811 | 8.001 | 0.794 | 0.026 | 0.830 | 0.897 |
| Ours-$\mathcal{G}^*_{mask}$ | 34.253 | 0.914 | 7.873 | 172.520 | 8.045 | 0.670 | 0.018 | 0.855 | 0.917 |
| **Ours-$\mathcal{G}_{free}$** | 34.425 | 0.934 | 7.031 | 176.630 | 8.562 | 0.630 | 0.014 | 0.883 | 0.923 |

*Table 2.* **Quantitative results on X-DubBench.** "Ref." is short for reference. Top three results are highlighted as first , second , and
third . LatentSync† denotes LatentSync retrained on the same 600-hour internet video corpus.

| | X-DubBench | | | | | | | | |
|---|---|---|---|---|---|---|---|---|---|
| | Visual Quality (Ref.) | | Visual Quality (No Ref.) | | | Lip Sync | Identity | | Generation |
| Method | FID ↓ | FVD ↓ | NIQE ↓ | BRISQUE ↓ | HyperIQA ↑ | Sync-C ↑ | CSIM ↑ | CLIPS ↑ | Success Rate ↑ |
| Wav2Lip | 19.330 | 631.589 | 6.908 | 48.397 | 35.667 | 5.087 | 0.738 | 0.805 | 62.95% |
| VideoReTalking | 17.535 | 341.951 | 6.392 | 43.112 | 44.826 | 5.126 | 0.684 | 0.793 | 59.09% |
| TalkLip | 21.262 | 550.658 | 6.284 | 38.990 | 34.311 | 3.213 | 0.739 | 0.724 | 70.45% |
| IP-LAP | 14.891 | 328.728 | 6.576 | 44.879 | 38.059 | 2.292 | 0.797 | 0.809 | 57.73% |
| Diff2Lip | 17.126 | 378.527 | 6.554 | 44.059 | 36.872 | 4.702 | 0.705 | 0.799 | 71.82% |
| MuseTalk | 17.519 | 294.312 | 6.552 | 43.778 | 42.335 | 2.205 | 0.672 | 0.753 | 60.00% |
| LatentSync | 13.602 | 265.057 | 6.113 | 39.154 | 41.654 | 6.282 | 0.801 | 0.812 | 59.77% |
| LatentSync† | 12.834 | 253.931 | 6.125 | 38.216 | 44.573 | 6.428 | 0.788 | 0.812 | 59.77% |
| Ours-$\mathcal{G}^*_{mask}$ | 10.824 | 224.893 | 5.920 | 36.840 | 48.120 | 6.514 | 0.814 | 0.818 | 66.05% |
| **Ours-$\mathcal{G}_{free}$** | 9.351 | 214.298 | 5.782 | 29.870 | 51.960 | 7.282 | 0.850 | 0.839 | 96.36% |

mask-free $\mathcal{G}_{free}$ enables precise lip editing with faithful identity preservation and robustness to spatiotemporal variations. Unlike mask-based methods that rely on human-face priors (e.g., landmarks) and often fail on stylized or non-human characters (marked "ERROR"), our mask-free video dubber implicitly localizes speech-relevant regions without mask heuristics, yielding stable performance across diverse character types and occlusions. Furthermore, by operating on the full input video with frame-wise alignment to the target output, $\mathcal{G}_{free}$ benefits from complete spatiotemporal context and remains free of identity or color drift even on videos longer than one minute.

**User study.** We further conduct a user study with 30 participants on 24 dubbing videos from different methods, collecting Mean Opinion Scores (MOS). Each video is rated on a 5-point Likert scale for realism, lip sync, identity preservation, and overall quality. As shown in Tab. 3, our method

*Table 3.* **User study results** of MOS with 95% confidence intervals. Top three results are highlighted as first , second , and third .

| Method | Realism ↑ | Lip Sync ↑ | Identity ↑ | Overall ↑ |
|---|---|---|---|---|
| Wav2Lip | 2.56±0.11 | 2.80±0.13 | 3.07±0.14 | 2.35±0.10 |
| VideoReTalking | 3.00±0.09 | 3.09±0.11 | 3.58±0.09 | 3.22±0.11 |
| TalkLip | 2.59±0.13 | 2.08±0.11 | 3.06±0.11 | 2.73±0.11 |
| IP-LAP | 2.74±0.09 | 2.49±0.11 | 3.62±0.11 | 3.09±0.11 |
| Diff2Lip | 2.63±0.11 | 2.91±0.13 | 3.22±0.13 | 2.62±0.12 |
| MuseTalk | 2.45±0.10 | 2.35±0.11 | 2.98±0.14 | 2.49±0.11 |
| LatentSync | 2.91±0.11 | 2.81±0.12 | 3.62±0.11 | 3.16±0.13 |
| Ours-$\mathcal{G}^*_{mask}$ | 4.28±0.07 | 3.87±0.09 | 4.02±0.12 | 4.48±0.08 |
| **Ours-$\mathcal{G}_{free}$** | 4.40±0.06 | 4.50±0.06 | 4.40±0.07 | 4.66±0.05 |

holds clear margins over existing baselines across all aspects. Moreover, our $\mathcal{G}_{free}$ surpasses $\mathcal{G}^*_{mask}$, particularly in identity consistency and lip sync, validating the bootstrapping effect that yields perceptually superior and higher-quality dubbing.

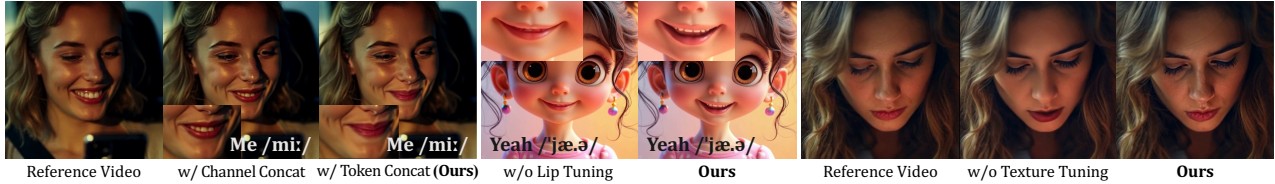

**Figure 4. Qualitative comparisons** across diverse scenarios. Lip-sync errors are marked with yellow, visual artifacts with blue, and lip leakage during silence with red. "ERROR" indicates runtime failure from missing 3DMM or landmarks despite best efforts. Our method exhibits robust performance with superior lip accuracy and identity consistency. Please 🔍**zoom in** for details.

**Figure 5. Ablations.** *Channel concat* (vs. our *token concat*) harms lip sync. Omitting *lip* or *texture* phases degrades sync or fidelity.

## 4.3. Ablation Study

We conduct ablations on two key components: 1) reference video injection mechanism, and 2) timestep-adaptive multi-phase learning, with results in Tab. 4 and Fig. 5.

For reference conditioning, replacing our frame-level token concatenation with channel concatenation causes a 12.5%

*Table 4.* **Ablation results** on HDTF dataset. Top two results are highlighted as first and second .

| Method | FID ↓ | Sync-C ↑ | LPIPS ↓ | CSIM ↑ |
|---|---|---|---|---|
| **Ours-$\mathcal{G}_{\text{free}}$ (full)** | 7.03 | 8.56 | 0.014 | 0.883 |
| w/ channel concat | 8.89 | 7.49 | 0.014 | 0.873 |
| w/ uniform $t$ | 18.52 | 3.85 | 0.125 | 0.592 |
| w/o lip tuning | 7.00 | 7.68 | 0.013 | 0.875 |
| w/o texture tuning | 8.26 | 8.56 | 0.018 | 0.847 |

drop in Sync-C, also visible as lip-shape errors in Fig. 5. Channel concatenation enforces rigid spatial fusion that conflicts with lip editing, while our token-based design uses self-attention to transfer identity without disturbing lips.

For training, replacing progressive multi-phase sampling with uniform timestep sampling, i.e., learning all noise levels at once, causes severe degradation and even divergence. Removing the lip phase reduces lip sync (–10.3%), with negligible gains in FID and LPIPS, while removing the texture phase weakens fidelity and identity (CSIM –4.1%). These results confirm the complementarity of our phases: the model sequentially learns global structure (high-noise), lip motion (mid-noise), and fine-grained texture (low-noise). This progressive decomposition facilitates learning by allowing the network to address distinct features step-by-step, rather than struggling to optimize all conflicting objectives simultaneously. Additional ablations on timestep selection and sensitivity analysis can be found in Appendix C.3.

## 5. Conclusion

In this paper, we introduce a generative bootstrapping paradigm to address the core challenge in visual dubbing: the scarcity of paired data differing only in lip motion. By repurposing a mask-guided model to synthesize high-fidelity pseudo-pairs, we finally bootstrap a robust mask-free video dubber. This design fundamentally eliminates mask-induced artifacts, such as boundary leakage and identity drift. The training is further bolstered by a timestep-adaptive multi-phase strategy that disentangles the optimization of global structure, lip motion, and fine-grained texture, ensuring high-fidelity output. Experiments on standard datasets and our challenging X-DubBench demonstrate that we achieve state-of-the-art results with superior robustness in complex, in-the-wild scenarios. Beyond visual dubbing, we believe this framework offers valuable insights for other conditional video editing tasks where paired supervision is scarce.

## Acknowledgements

This work is supported by National Natural Science Foundation of China (62076144) and Shenzhen Science and Technology Program (JCYJ20220818101014030).

## Impact Statement

This work advances the field of talking head generation and visual dubbing by introducing a highly generalizable editing paradigm. While this technology holds great promise for applications in education, virtual assistants, and multilingual media, it also necessitates careful consideration of its societal implications. The ability to synthesize realistic audio-visual content raises concerns regarding identity impersonation and the spread of synthetic media. We stress the importance of transparency, such as clear labeling of AI-generated content, and support ongoing efforts in the research community to develop robust detection mechanisms to mitigate potential misuse.

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

# A. Details of Our DiT-Based Data Generator and Video Dubber

## A.1. Preliminary of Flow-Matching-Based DiT Models

We adopt a pre-trained T2V DiT model as the backbone for both stages (Hong et al., 2022; Kong et al., 2024; Wan et al., 2025). It follows a latent diffusion paradigm with a 3D causal Variational Auto-Encoder (VAE) (Kingma & Welling, 2013) for video compression and a DiT (Peebles & Xie, 2023) for sequence modeling. Each DiT block interleaves 2D (spatial) self-attention, 3D (spatio-temporal) self-attention, text cross-attention, and feed-forward networks (FFN). Training follows standard flow matching (Esser et al., 2024; Lipman et al., 2022) with the forward process:

$$z_t = (1 - t) z_0 + t \epsilon, \quad \epsilon \sim \mathcal{N}(0, I), \tag{4}$$

and a v-prediction objective to predict $v = \epsilon - z_0$ conditioned on $c$:

$$\mathcal{L}_{\text{FM}}(\theta) = \mathbb{E}_{z_0, \epsilon, t} \left[ \left\| v_\theta(z_t, t, c) - v \right\|_2^2 \right]. \tag{5}$$

## A.2. Adaptation of Text Cross-Attention Mechanism

To effectively adapt the pre-trained backbone, originally designed for text-to-video generation, for the visual dubbing task, we implement a specific strategy for handling text cross-attention. During training, we utilize Qwen2.5-VL (Bai et al., 2025) to generate coarse captions for the target real videos, which serves to preserve the backbone's generative priors. However, to ensure the model primarily relies on the provided visual context and audio signals rather than textual descriptions, we apply a high dropout rate of 70% to these text conditions. Architecturally, the text embeddings interact exclusively with the noised target tokens via cross-attention, while the reference tokens remain unaffected. For inference, to maintain practical convenience without requiring user-provided captions, we employ an empty string as the positive prompt. Additionally, we leverage Classifier-Free Guidance (CFG) with standard negative prompts (e.g., "Blurry, deformed, low quality, distorted...") to suppress artifacts and ensure high-fidelity generation.

## A.3. Details of Our Mask-Based Data Generator

**Mask Setting.** Previous mask-based dubbing methods typically employ either half-face rectangular masks derived from smoothly varying bounding boxes (Prajwal et al., 2020; Cheng et al., 2022; Wang et al., 2023; Zhang et al., 2023) or fixed irregular-shaped masks applied to affine-transformed facial crops (Guan et al., 2023; Li et al., 2024). However, the former's size variations often induce lip motion information leakage, causing models to learn lip movements from visual occlusion changes rather than the conditional speech (i.e., shortcut learning). The latter constrains jaw movement, hindering the generation of pronounced mouth shapes such as wide-open expressions. Furthermore, both masking strategies often aggressively occlude the background context surrounding the face. This deprives the model of necessary spatial reference, frequently resulting in visible boundary artifacts or inconsistencies in the background after inpainting.

Instead, we utilize frame-wise 3D Morphable Models (3DMM) (Retsinas et al., 2024) to derive precise facial masks. Specifically, we project the facial mesh while fixing the jaw opening parameter to a maximum of 0.4, keeping all other pose and expression coefficients unchanged. After retaining the lower-half facial region, we further apply a morphological dilation to slightly cover mouth-boundary and jaw-adjacent regions. By using this dilated lower-half mask, we minimize the loss of spatiotemporal context (e.g., background dynamics) to reduce inpainting artifacts, while effectively preventing ground-truth lip-shape leakage.

**Audio Conditioning.** Audio features are extracted using the Whisper (Radford et al., 2023) encoder and then injected via an audio cross-attention layer placed after text cross-attention. Since visual tokens and audio features have different temporal resolutions (1 video token frame corresponds to 8 audio-feature frames, i.e., 1:8), for each video frame, we select the corresponding audio feature frames according to the timestamp, together with neighboring frames, forming a temporal window of size $n = 16$. This yields audio tokens $h_a \in \mathbb{R}^{(b \times f) \times n \times c}$, while video tokens are reshaped into $h_V \in \mathbb{R}^{(b \times f) \times (h' \times w') \times c}$, where $h' \times w'$ denotes the visual spatial size after patchification. Frame-wise cross-attention is then performed between the two modalities, where video tokens serve as queries and audio tokens as keys and values. Formally,

$$\text{Attn}(Q_V, K_A, V_A) = \text{softmax}\left( \frac{Q_V K_A^\top}{\sqrt{d}} \right) V_A, \quad Q_V = h_V W_Q^V, \ K_A = h_a W_K^A, \ V_A = h_a W_V^A. \tag{6}$$

**Reference Conditioning.** Reference frames $\{I_{\text{ref}}^i\}_{i=1}^N$ are sampled from a different segment of the same video during training to prevent lip-shape leakage, while at inference from the target segment to provide visual cues under a similar head pose.

### A.4. Details of Our Mask-Free Video Dubber

**3D Rotary Position Embedding (RoPE).** 3D RoPE is adopted in 3D self-attention of the DiT backbone to distinguish spatial-temporal positions, which we keep unchanged for target tokens. For reference tokens, inspired by Tan et al. (2024), we adapt RoPE to be temporally-aligned but spatially-shifted. Specifically, a reference token located at $(i, j, k)$, where $i$, $j$, and $k$ denote the height, width, and temporal indices, is mapped to $(i + h', j + w', k)$, with $(h', w', f')$ the spatial–temporal sizes after patchification. This design provides two benefits: 1) Temporal alignment enables frame-wise consistency preservation of dynamic attributes such as background and head poses; 2) Spatial shifting avoids direct overlap that could distort lip movements, and instead encourages the model to capture spatially misaligned yet correlated features like identity information.

## B. Details of Data Construction Strategies

### B.1. Short-Term Segment Processing

During the video generator inference with a single reference frame, we observe that denoising a long clip of 77 frames (matching the setting used by the backbone and the final video dubber) in one pass causes noticeable texture and color drift in the tail frames relative to the first; the drift resets at the first frame of the next clip (see Fig. 6). Therefore, *under a single-reference regime, we conclude that single-pass denoising over long clips is detrimental to identity preservation.* We hypothesize two contributing factors: 1) the reference frame is anchored at the first position, so later frames become distant in the RoPE index space, amplifying identity drift; and 2) long clips naturally accumulate larger head motion and spatiotemporal changes, which a single reference frame cannot fully constrain.

To mitigate this, when constructing pseudo pairs with the generator, we adopt short-segment training and inference: we generate clips of 25 frames and bridge adjacent clips with 5 motion frames, then concatenate them to form videos longer than 77 frames for supervising the mask-free video dubber. This short-segment strategy enhances identity preservation, while any slight sacrifice in lip sync accuracy remains within our design guidelines, as shown in Tab. 5.

| Frame 1
(**First** frame in segmentation 1) | Frame 77
(**Last** frame in segmentation 1) | Frame 78
(**First** Frame in segmentation 2) |
| --- | --- | --- |

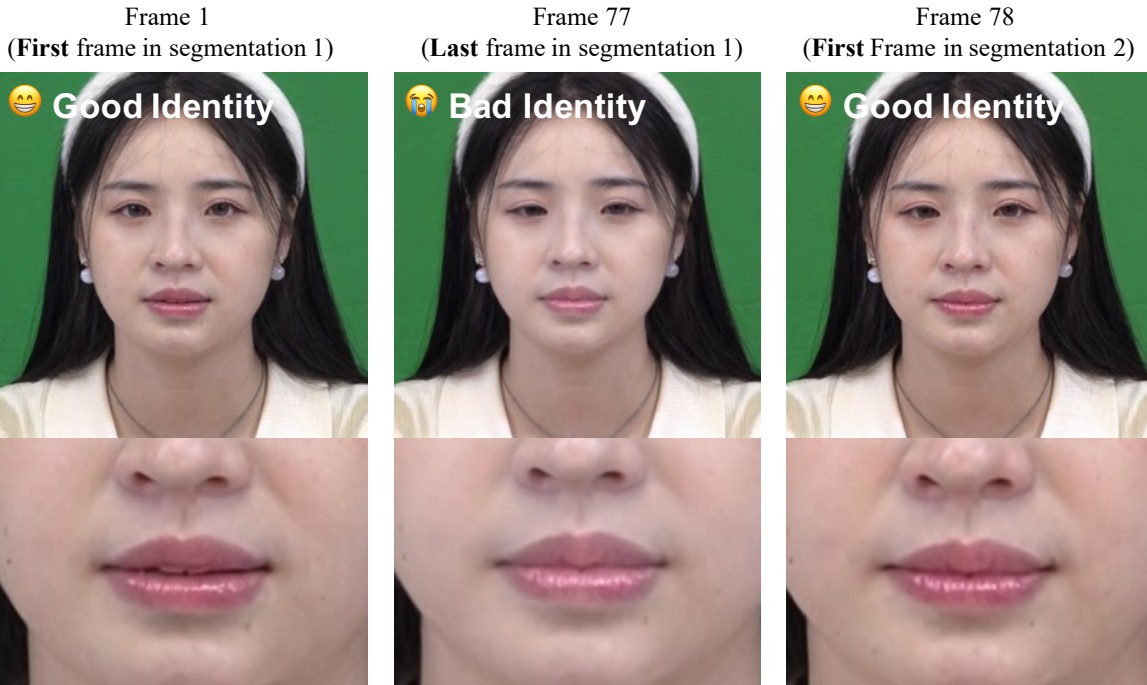

*Figure 6.* Example of intra-segment identity drift.

Furthermore, this strategy offers a significant computational advantage. Since the computational complexity of the attention mechanism grows quadratically with sequence length ($O(T^2)$), generating a long sequence in a single pass is computationally expensive. By slicing the video into shorter segments, the generation cost scales linearly with the number of segments. Even with the overhead of overlapping frames, this approach to some extent accelerates the data construction process.

*Table 5.* Quantitative results comparing long-term vs. short-term processing strategies.

| Method | Sync-C (Lip Sync) ↑ | CSIM (Identity Preservation) ↑ |
|---|---|---|
| Long-clip (77 frames) | 7.983 | 0.842 |
| Short-segment (25 frames, +5 overlap) | 7.841 | 0.867 |

## B.2. Multiple Reference Frames for the Mask-Based Data Generator

Building upon the 25-frame short-segment setting established above, we further investigate the impact of the number of reference frames ($N$) on the quality of the constructed data. We hypothesize that providing additional visual context can assist the mask-based generator in better maintaining identity and visual fidelity during inpainting.

We conducted experiments on the HDTF dataset with varying numbers of reference frames ($N = \{1, 2, 4, 6\}$). As presented in Tab. 6, increasing the number of reference frames consistently improves performance across all metrics. Specifically, we observe a steady decrease in FID and LPIPS, indicating better visual quality and perceptual similarity, alongside an increase in CSIM, reflecting improved identity preservation.

However, the performance gains begin to saturate beyond $N = 4$. The improvement from $N = 4$ to $N = 6$ is marginal (e.g., LPIPS remains constant at 0.020), while the computational cost for encoding reference features continues to increase. Therefore, to strike an optimal balance between generation quality and data construction efficiency, we select $N = 4$ as our default setting.

*Table 6.* Impact of the number of reference frames on the mask-based generator (evaluated on the HDTF dataset). **Bold** indicates the best, while underline indicates the second best.

| # Ref. Frames | FID ↓ | LPIPS ↓ | CSIM ↑ |
|---|---|---|---|
| 1 | 8.01 | 0.025 | 0.892 |
| 2 | 7.80 | 0.022 | 0.895 |
| 4 | 7.72 | 0.020 | **0.904** |
| 6 | **7.70** | **0.020** | 0.903 |

## B.3. Mask Processing with Occlusion Handling

To enhance the robustness of our generator against occlusions, namely, to maintain consistency with the original video's occlusion patterns and thereby facilitate the editor's ability to naturally inherit them, we introduce an occlusion-handling pipeline. First, a vision–language model (VLM) (Bai et al., 2025) is prompted per video with: *"Does any object occlude the person's face? If yes, output **only** a concise description of the object(s). If no, output nothing."* The returned object phrase(s) are then passed to SAM 2 (Ravi et al., 2024) to segment candidate occluders, yielding an occlusion mask $M_{\text{occ}}$. We apply a light manual screening step to remove severely erroneous segmentations.

Finally, we compose the occlusion-aware mask with the original inpainting mask. Let $M_{\text{face}}$ be the face mask (foreground 1, background 0), and $M_{\text{occ}}$ the occluder mask (1 on occluding objects). The visible-face mask is

$$M_{\text{vis}} = M_{\text{face}} \wedge \neg M_{\text{occ}},$$

and the inpainting mask (where *0* indicates regions to inpaint in our implementation) is

$$M_{\text{inp}} = \neg M_{\text{vis}} = \neg M_{\text{face}} \vee M_{\text{occ}},$$

where $\wedge$, $\vee$, and $\neg$ denote logical AND, OR, and NOT, respectively. As illustrated in Fig. 7, $M_{\text{inp}}$ excludes occluders while preserving non-occluded facial areas for inpainting.

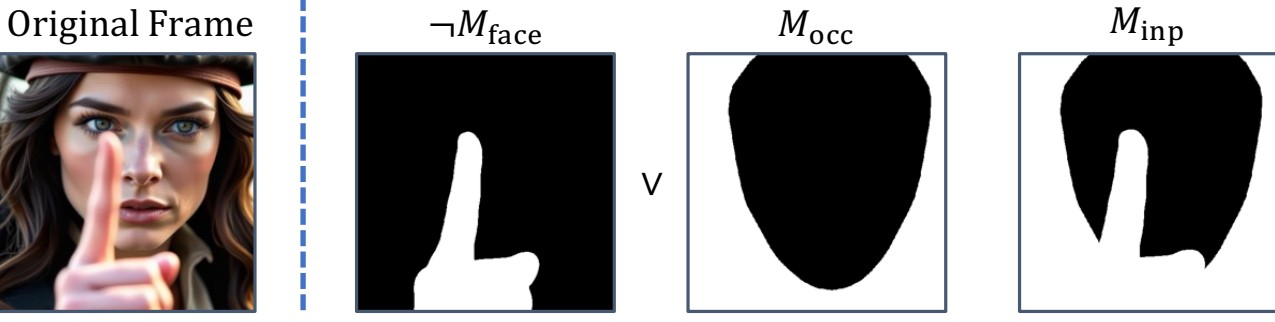

*Figure 7.* Example of mask processing with occlusion handling.

While our occlusion annotations can be incomplete or noisy, and using occlusion masks may introduce slight blur near mask boundaries and occasional lip degradation, as shown in the main text, the pipeline still supplies *paired and coherent* references that preserve the scene's occlusion patterns. This supervision encourages the final mask-dubber to model occlusion–face interactions automatically, enabling robust handling of occlusions without labor-intensive manual intervention.

### B.4. Lighting Augmentation

To enhance the robustness of our mask-free video dubber against challenging lighting conditions, we leverage a video relighting method (Bian et al., 2025) to augment our paired training data. As illustrated in Fig. 8, we apply identical relighting effects to both the original target video $V_{real}$ and its synthetic companion $V_{syn}$.

This synchronized processing ensures that the synthetic video input remains fully frame-aligned with the target, allowing the model to learn to directly inherit global lighting structures from the input. Our augmentation strategy includes both static adjustments (varying chromaticities and intensities) and dynamic effects, where light source properties change continuously across frames. To maintain high visual fidelity, we primarily apply this augmentation to high-quality studio footage and restrict it to approximately 5% of the total training dataset. This conservative ratio prevents potential relighting artifacts from degrading the model, while effectively improving generalization to in-the-wild lighting dynamics.

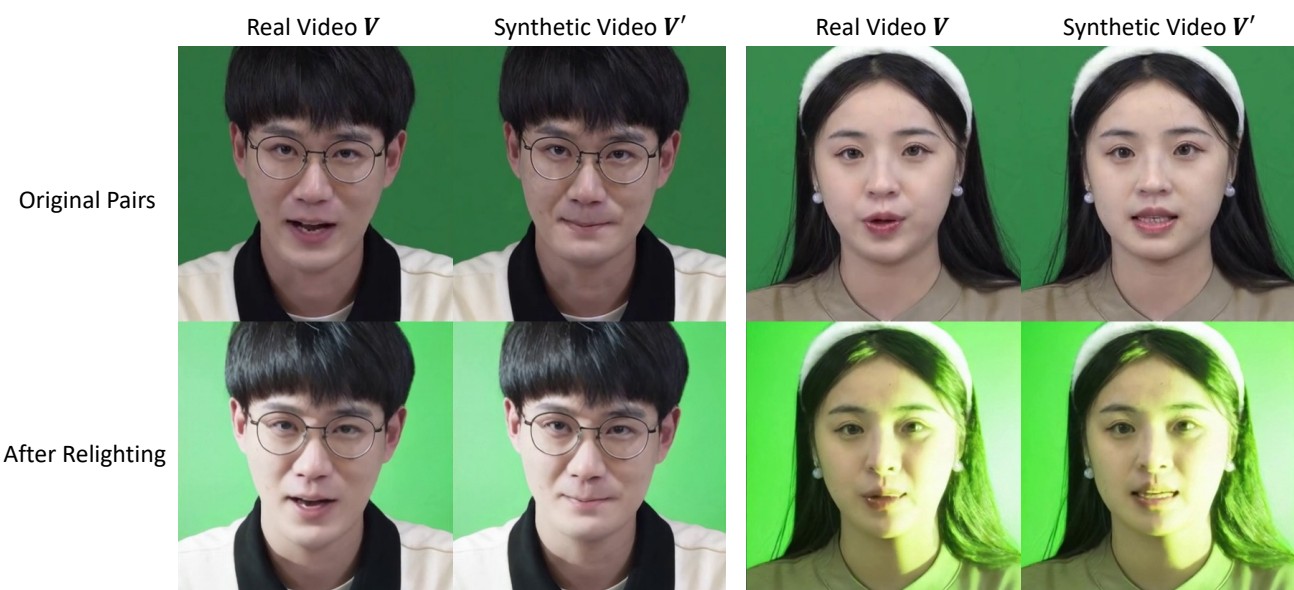

*Figure 8.* **Visualizing the lighting augmentation strategy.** We apply identical static and dynamic relighting effects (e.g., changing color, intensity, and direction) to both the generated contextual reference and the target video. This ensures the mask-free video dubber learns to utilize lighting cues from the complete input video even under varying illumination conditions.

### B.5. Post-Processing

Similar to Zhong et al. (2023), we use a Gaussian-smoothed face mask in post-processing to composite the data generator's facial region back onto the original frames, mitigating minor background and boundary artifacts. Concretely, we blur the binary face mask $M_{\text{face}}$ with a Gaussian kernel to obtain $\tilde{M}_{\text{face}} \in [0, 1]$, and perform per-frame alpha blending:

$$\boldsymbol{V}_{\text{post}} = \tilde{M}_{\text{face}} \odot \boldsymbol{V}_{\text{gen}} + \left(1 - \tilde{M}_{\text{face}}\right) \odot \boldsymbol{V}_{\text{orig}},$$

where $\odot$ denotes element-wise multiplication. This feathered composition keeps backgrounds consistent while preserving sharp facial edits, yielding training pairs with background-aligned context and helping the editor learn background-consistent editing behavior.

### B.6. Quality Filtering

To maintain identity consistency, enforce distinct lip shapes, and safeguard visual quality, we apply the following filters to each synthetic-original pair.

**Identity Similarity Filtering.** We use ArcFace (Deng et al., 2019) to compute cosine similarity between the synthetic and original videos. As a reference, the mean within-speaker similarity across different real segments is 0.812, which is conservative given their differing head motions. Since our paired videos share identical head motion, we adopt a stricter threshold of 0.85 and discard pairs below this value to prevent identity drift.

**Lip-Shape Distinction Filtering.** After aligning faces to a canonical template using the Umeyama algorithm following (Deng et al., 2019), we measure the normalized landmark distance over the mouth region between the original and synthetic videos. To further capture mouth-opening differences that may be under-emphasized by landmark distance alone, we additionally compute the frame-wise mouth aspect ratio (MAR), defined as the vertical lip opening normalized by the mouth-corner width. We then compare the temporal MAR trajectories between the original and synthetic videos. To ensure sufficient lip-shape variation, we reject pairs with a mouth-region landmark distance below 1.0 or insufficient MAR difference. This excludes pairs with overly similar lip shapes and reduces leakage propagation to the subsequent mask-free training stage.

**Visual Quality Filtering.** To further safeguard visual quality and remove synthetic companions containing noticeable artifacts, we additionally assess each generated clip using a multimodal video-quality model (Han et al., 2025). Each video is rated on six aspects including image fidelity, aesthetic appeal, temporal stability, motion smoothness, background consistency, and subject consistency, under a 5-point scoring scheme where 1 means very poor while 5 means excellent. We compute the average score across all six dimensions for each clip and retain only those with a mean score above 4.0, ensuring that only high-quality, artifact-free companion videos are included in the final training set.

### B.7. 3D Talking Head Rendering Data

We leverage Unreal Engine to generate high-quality dubbing pairs. Initially, we acquire the 3D motion representation, which comprises ARKit-based facial expressions and 3D degree-of-freedom (3DOF) head poses. For each dataset entry containing speech audio and 3D motion representation $(A, M)$, we randomly select another entry $(A', M')$, and replace the speech-correlated coefficients in M with those from $M'$ to form $M_{dub}$. Both the original dataset entry and its corresponding dubbed version are rendered as follows:

$$\begin{aligned} V &= \mathbf{R}(A, M, I), \\ V_{\text{dub}} &= \mathbf{R}(A', M_{\text{dub}}, I), \end{aligned} \tag{7}$$

where $\mathbf{R}$ denotes the Unreal Engine rendering pipeline (following (Chen et al., 2025a)) and $I$ represents the Unreal Engine MetaHuman avatar. To ensure data diversity, we create multiple avatars; however, it is important to note that the same avatar is used for each individual dubbing pair. Ultimately, we collect approximately 10 hours of 3D-rendered dubbing pairs in addition to the pairs generated by our DiT-based data generator. These rendered pairs provide strictly aligned head motion, environment, and perfectly matched identity, which further enables the mask-free video dubber to focus on speech-related lip edits while preserving all other visual cues. Rendering examples are shown in Fig. 9.

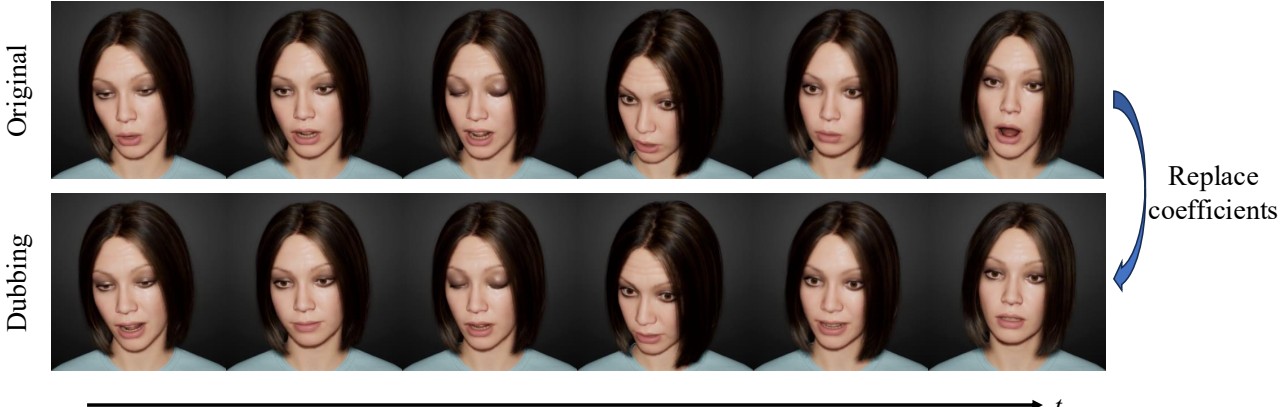

*Figure 9.* Example of aligned rendered video pairs.

## C. Details of Timestep-Adaptive Multi-Phase Learning

### C.1. Derivation of Eq. 3: Timestep-Constrained Single-Step Denoising

Given the forward diffusion process as in Eq. 4 and the v-prediction objective $v = \epsilon - z_0$, we derive the single-step denoising formula for pixel-level supervision during training, avoiding excessive computational overhead.

From Eq. 4, we can rearrange to obtain:

$$z_0 = \frac{z_t - t\,\epsilon}{1 - t}. \tag{8}$$

Since $v = \epsilon - z_0$, we have $\epsilon = v + z_0$. Substituting and solving for $z_0$:

$$z_0 = \frac{z_t - t(v + z_0)}{1 - t} = \frac{z_t - tv - tz_0}{1 - t}$$
$$(1 - t)z_0 = z_t - tv - tz_0 \tag{9}$$
$$z_0 = z_t - tv.$$

During inference, we use the predicted velocity $\hat{v}$ instead of the true $v$, yielding:

$$\hat{z}_0 = z_t - t\hat{v}. \tag{10}$$

Alternatively, we can express this as:

$$\hat{z}_0 = z_0 + t(v - \hat{v}), \tag{11}$$

which shows the reconstruction error depends on the velocity prediction error scaled by $t$.

However, when the timestep $t$ approaches 1 (high noise levels), the velocity prediction error $(v - \hat{v})$ tends to be amplified when multiplied directly by $t$. This leads to distorted reconstructions $\hat{x}_0$ that result in inaccurate and unstable gradients for lip-sync and identity losses.

To address this, we introduce a truncated effective timestep $\tau$ for the single-step reconstruction:

$$\hat{x}_0 = \mathcal{D}(z_0 + (v - \hat{v}) \cdot \tau), \tag{12}$$

where $\tau = \min(t, t_{\text{thres}})$. Importantly, this truncation is applied *exclusively* in the denoising computation for loss supervision. The model's forward pass remains conditioned on the actual timestep $t$, enabling it to learn essential global structure and lip movement patterns even in high-noise regions. By capping the scaling factor at $t_{\text{thres}}$ (set to 0.6 in our experiments), we stabilize the training objectives without restricting the model's perception of the noise level.

**Justification of the Truncation Threshold.** We empirically set $t_{\text{thres}} = 0.6$ based on both qualitative and quantitative observations. Qualitatively, single-step denoising reconstructions remain sufficiently clear and reliable for pixel-level

supervision when the effective timestep is below approximately 0.6, whereas reconstructions at higher noise levels become increasingly blurry and unstable. Quantitatively, we measure the MSE between clean frames and their single-step denoising reconstructions across different noise levels. As shown in Fig. 10, without truncation, the reconstruction error increases noticeably beyond $t \approx 0.6$, indicating that the resulting pixel-level losses would provide less reliable gradients. With the proposed truncation, the effective reconstruction error is bounded within the low-error region, preventing noisy gradients from destabilizing the SyncNet and identity supervision. Therefore, $t_{\text{thres}} = 0.6$ provides a practical trade-off: it preserves useful denoising supervision in mid- and low-noise regions while avoiding unreliable high-noise reconstructions.

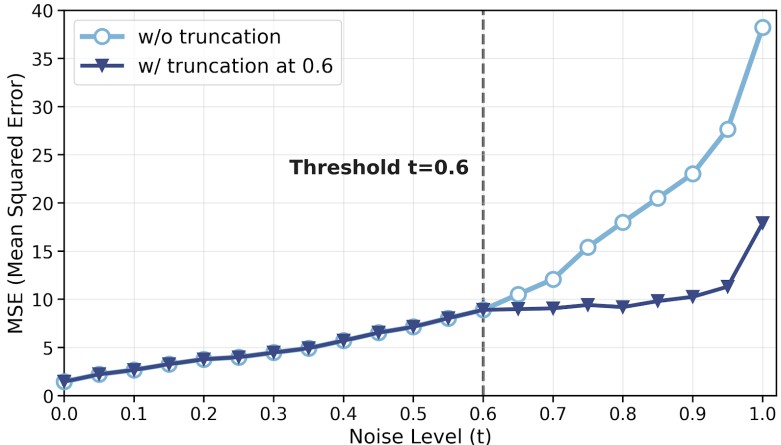

*Figure 10.* **Justification of the denoising truncation threshold.** We compute the MSE between clean frames and single-step denoising reconstructions across noise levels. Without truncation, the reconstruction error increases noticeably beyond $t \approx 0.6$, where denoised frames become less reliable for pixel-level supervision. Applying $t_{\text{thres}} = 0.6$ bounds the effective reconstruction error within the low-error region and stabilizes training.

### C.2. SyncNet Supervision

For lip-sync tuning, we adopt a SyncNet (Chung & Zisserman, 2016) comprising a visual encoder $S_V$ and an audio encoder $S_a$ to discriminate temporal alignment between video and audio clips. The lip-sync loss is defined as:

$$\mathcal{L}_{\text{sync}} = \text{CosSim}\big(S_V(\hat{\boldsymbol{x}}_0^{[f:f+8]}),\ S_a(\boldsymbol{a}^{[f:f+8]})\big). \tag{13}$$

This loss is combined with $\mathcal{L}_{\text{mFM}}$ defined in Eq. 1 in a weighted sum to train the lip-sync LoRA:

$$\mathcal{L}_{\text{total}} = (1 + \lambda_{\text{face}} \cdot \boldsymbol{M}_{\text{face}} + \lambda_{\text{lip}} \cdot \boldsymbol{M}_{\text{lip}}) \odot \mathcal{L}_{\text{FM}} + \lambda_{\text{sync}} \cdot \mathcal{L}_{\text{sync}}. \tag{14}$$

### C.3. Details of Parameter Choices for Multi-Phase Learning

Based on the well-established observation that diffusion models process information hierarchically (Peng et al., 2025; Zhang et al., 2025; Meng et al., 2025), we first heuristically partition the noise schedule (timestep $t$) into three functional regions: high-noise for global structure, mid-noise for lip motion, and low-noise for detailed texture. This initial design is then finalized via the quantitative experiments.

**Determination of Timestep Shifting $\alpha$ for Training.** To determine the optimal $\alpha$ values that allow the editing model to efficiently learn decoupled information in each phase, and also to maximize the impact of the lip-sync and identity losses without degrading overall quality, we conduct an ablation study on the specific $\alpha$ settings. The experiments for the mid- and low-noise phases are conducted sequentially, each building upon the optimal parameter choice from the preceding stage.

The results are presented in Tab. 7. For the high-noise phase focused on global structure, we find that performance is relatively insensitive to $\alpha$ values above 3.0, with the model stably converging to high visual quality with minimal changes in FID and LPIPS. This is consistent with findings in (Esser et al., 2024). We finally select $\alpha = 5.0$ to minimize overlap with the mid-noise phase.

For the mid-noise tuning phase, an overly large $\alpha$ (e.g., 3.0) degrades overall visual quality, while a low value (e.g., 0.8) diminishes the effectiveness of the lip-sync loss. The model's performance is relatively stable within the intermediate range. We therefore select $\alpha = 1.5$ as the best balance between effective lip modification and preserving visual quality.

Finally, for the low-noise phase, a larger $\alpha$ (e.g., 0.8) disrupts the previously learned lip shapes. Smaller values are more stable, and based on our results, we select $\alpha = 0.2$. This choice allows the texture tuning to maximize its enhancement of visual quality while minimizing any negative impact on the already-learned lip motion.

**Determination of Timestep Tntervals for LoRA Expert Activation for Inference.** Similarly, we conduct an ablation study on the activation timestep ranges for the two trained LoRA experts during inference to determine the optimal phase boundaries. Note that this experiment uses the LoRA checkpoints trained with the optimal $\alpha$ values determined previously.

The results are shown in Tab. 8. First, for both experts, naively activating them across the entire denoising process ($t \in [0.0, 1.0]$) leads to a severe degradation in either visual quality or lip-sync, as this forces them to operate in timestep regions they rarely encountered during training.

For the lip LoRA expert trained in the mid-noise phase, activating it too early ($t \rightarrow 1$) conflicts with global visual quality and causes flickering artifacts around the mouth. Activating it too late ($t \rightarrow 0$) fails to sufficiently enhance lip sync. For the texture LoRA expert trained in the low-noise phase, activating it too early degrades the quality of the lip motion. Therefore, to balance these trade-offs, we select the non-overlapping ranges of $t \in [0.4, 0.8]$ for the lip LoRA and $t \in [0.0, 0.3]$ for the texture LoRA.

*Table 7.* **Ablation study on the timestep shifting parameter $\alpha$ for each training phase on the HDTF dataset. Bold** indicates the best within a phase, while underline indicates the second best.

| Phase | $\alpha$ | Approximate Peak of $t$ | FID $\downarrow$ | LPIPS $\downarrow$ | Sync-C $\uparrow$ | Choices |
|---|---|---|---|---|---|---|
| | 5.0 | 0.921 | 8.25 | **0.017** | **7.68** | ✓ |
| **High-noise** | 4.0 | 0.899 | **8.24** | 0.018 | 7.64 | |
| | 3.0 | 0.861 | 8.31 | 0.020 | 7.65 | |
| | 3.0 | 0.861 | 10.52 | 0.021 | 8.47 | |
| **Mid-noise** | 2.0 | 0.777 | 8.39 | **0.017** | 8.50 | |
| | 1.5 | 0.684 | **8.26** | 0.018 | **8.56** | ✓ |
| | 0.8 | 0.392 | 8.31 | 0.018 | 7.21 | |
| | 0.8 | 0.392 | 7.25 | 0.015 | 7.98 | |
| **Low-noise** | 0.4 | 0.172 | **7.00** | 0.015 | 8.43 | |
| | 0.2 | 0.079 | 7.03 | **0.014** | **8.56** | ✓ |

*Table 8.* **Ablation study on the activation timestep ranges for LoRA experts during inference on the HDTF dataset. Bold** indicates the best, while underline indicates the second best.

| LoRA Expert | Timestep Range | FID $\downarrow$ | LPIPS $\downarrow$ | Sync-C $\uparrow$ | Choices |
|---|---|---|---|---|---|
| | [0.0, 1.0] | 9.24 | 0.028 | 7.92 | |
| **Lip** | [0.6, 1.0] | 8.52 | 0.020 | **8.61** | |
| | [0.4, 0.8] | **7.03** | **0.014** | 8.56 | ✓ |
| | [0.2, 0.6] | 7.26 | 0.016 | 8.03 | |
| | [0.0, 1.0] | **6.95** | 0.015 | 6.74 | |
| **Texture** | [0.1, 0.4] | 7.54 | 0.016 | 7.99 | |
| | [0.0, 0.3] | 7.03 | **0.014** | **8.56** | ✓ |

# D. Other Implementation Details

We conduct experiments using a $\sim$1B-parameter T2V model on 32 A100 GPUs, with face-centered videos at $512 \times 512$ resolution and 25 fps. For the data generator, we conduct training for $\sim$15k steps on 600 hours of internet audio-video data, sampling 25 frames with lr=1e-5 and batch size 256, which takes $\sim$1 day. Using the trained data generator, we then synthesize the contextual video pairs, a one-time data preparation step that takes $\sim$2 days. After inference and curation, we obtain 400 hours of video pairs, totaling 800 hours. For the mask-free video dubber, we begin with full-parameter training for $\sim$4k steps on 77-frame samples with lr=1e-5, batch size 256, and timestep shift $\alpha = 5.0$, followed by LoRA expert training for $\sim$1k steps each with lr=5e-6 and batch size 64; the entire training process for the video dubber takes $\sim$0.5 days. To reduce computational cost, we decode 4 tokens into 13-frame segments for pixel-level loss computation. Timestep shifts

are set to $\alpha = 1.5$ for the lip expert and $\alpha = 0.2$ for the texture expert. Loss weights are set as $\lambda_{\text{face}} = \lambda_{\text{lip}} = 0.6$ for masks, and 0.05 for SyncNet, CLIP, and ArcFace loss.

## E. Inference Time and Computational Cost

Inference with our mask-free video dubber $\mathcal{G}_{\text{free}}$ requires approximately 30 GB of VRAM and fits comfortably on a single A100 GPU. With 50 denoising steps, $\mathcal{G}_{\text{free}}$ takes about 1 minute to process a 3-second, 25 fps video at $512 \times 512$ resolution.

To provide a comprehensive performance profile, we compare our mask-free video dubber $\mathcal{G}_{\text{free}}$'s inference time against some representative methods: a GAN-based dubbing model (Wav2Lip), a diffusion-based dubbing model (LatentSync), and a large-scale single-image animation model (MultiTalk). All diffusion-based methods are benchmarked with 50 denoising steps for a fair comparison, as shown in Tab. 9.

*Table 9.* Inference time comparison on a single A100 GPU. All diffusion models use 50 steps. The task is to process a 3-second, 25 fps video.

| Method | Wav2Lip | LatentSync | MultiTalk | **Ours-$\mathcal{G}_{\text{free}}$** |
|---|---|---|---|---|
| Model Type | GAN | Diffusion (UNet) | Diffusion (DiT) | Diffusion (DiT) |
| Parameters | ~36M | ~816M | ~14B | ~1.5B |
| Inference Time | ~1s | ~30s | ~1800s (30 min) | ~60s (1 min) |

The results in Tab. 9 show a clear quality-efficiency trade-off. As expected, our DiT-based mask-free video dubber is slower than lightweight GAN-based methods like Wav2Lip, but its inference speed is comparable to other diffusion-based dubbing methods such as LatentSync. Crucially, our method achieves this comparable speed while delivering substantially better lip-sync accuracy and visual quality. Furthermore, when compared to large-scale animation models, our 1.5B parameter mask-free video dubber achieves an overall quality comparable to the 14B MultiTalk model, yet requires only a fraction of its inference time and parameter size. This demonstrates that a task-specific design for visual dubbing is more cost-effective than simply scaling up to a much larger, general-purpose video generation backbone.

Finally, our method can be significantly accelerated. Benefiting from paired data and the utilization of full frame-aligned video inputs during inference, the early, high-noise denoising steps primarily involve inheriting global structure and low-frequency components directly from the original video. This allows us to safely reduce the total number of inference steps to 25 (mainly by pruning the early and late stages) without noticeable quality degradation. Combined with lightweight acceleration techniques such as sequence parallelism and test-time caching (*e.g.*, TeaCache (Liu et al., 2024)), we can shorten the inference time to approximately 25 seconds for a 3-second clip, substantially mitigating practical deployment limitations.

**Quality under acceleration.** We further quantify the effect of the above acceleration strategy on generation quality. Specifically, we compare the original inference setting with 50 denoising steps against the accelerated setting with 25 denoising steps and TeaCache (Liu et al., 2024). As shown in Tab. 10, acceleration introduces only marginal degradation in visual quality while keeping lip synchronization almost unchanged. On HDTF, Sync-C remains identical after acceleration, and FID/LPIPS increase only slightly. On X-DubBench, the accelerated model also maintains comparable FID and Sync-C, with only a minor increase in LPIPS. These results demonstrate that our acceleration strategy substantially improves practical efficiency while preserving the main quality advantages of our mask-free video dubber.

*Table 10.* **Quality comparison before and after inference acceleration.** Original $\rightarrow$ accelerated, where acceleration uses 25 denoising steps with TeaCache.

| Dataset | FID $\downarrow$ | LPIPS $\downarrow$ | Sync-C $\uparrow$ |
|---|---|---|---|
| HDTF | $7.03 \rightarrow 7.18$ | $0.014 \rightarrow 0.016$ | $8.56 \rightarrow 8.56$ |
| X-DubBench | $9.35 \rightarrow 9.41$ | $0.023 \rightarrow 0.029$ | $7.28 \rightarrow 7.25$ |

## F. Additional Analyses of the Inpainting-to-Editing Paradigm

### F.1. Ablation on Generative Bootstrapping Paradigm vs. Training Strategy

To clearly disentangle the contributions of our two core components (the generative bootstrapping paradigm (using constructed paired data) and the timestep-adaptive multi-phase learning strategy), we conduct a crucial ablation study. We compare four settings, varying the paradigm (inpainting vs. editing) and the training strategy (single-phase vs. multi-phase).

*Table 11.* Ablation study disentangling the contributions of the paradigm (inpainting vs. editing) and the training strategy (single-phase vs. multi-phase). Best results are in **bold**.

| | Method | Paradigm | Training Strategy | FID $\downarrow$ | LPIPS $\downarrow$ | Sync-C $\uparrow$ | Remarks |
|---|---|---|---|---|---|---|---|
| ① | $\mathcal{G}_{mask}^{*}$ | Inpainting | Single-Phase (Uniform) | 7.87 | 0.018 | 8.05 | |
| ② | $\mathcal{G}_{mask}^{*}$ | Inpainting | Multi-Phase | 7.92 | 0.018 | 8.19 | |
| ③ | $\mathcal{G}_{free}$ | Editing | Single-Phase (Uniform) | 18.52 | 0.125 | 7.68 | Not converged. |
| ④ | $\mathcal{G}_{\mathbf{free}}$ | **Editing** | **Multi-Phase** | **7.03** | **0.014** | **8.56** | |

The results in Tab. 11 lead to two key findings. First, the primary performance gain stems from the paradigm shift enabled by the constructed paired data. Applying multi-phase learning to the inpainting-based $\mathcal{G}_{mask}^{*}$ (② vs. ①) yields negligible gains, and its performance remains far below that of our final mask-free video dubber (④). This demonstrates that the training strategy alone cannot overcome the fundamental limitations of the mask-based inpainting paradigm, which lacks frame-aligned visual context and is constrained by the explicit mask.

Second, the multi-phase learning strategy is an essential enabler for the mask-free video dubber $\mathcal{G}_{free}$, but not the $\mathcal{G}_{mask}^{*}$. Mask-based $\mathcal{G}_{mask}^{*}$ converges well with uniform sampling (①) as its inpainting task is straightforward generation. The mask-free editing model, in contrast, must balance the conflicting objectives of inheriting structure, editing lips, and preserving texture. A standard single-phase approach mixes these signals and causes training to collapse (③), whereas our multi-phase strategy disentangles them, enabling stable and effective training (④). In summary, the paradigm shift is the primary source of improvement (bootstrapping effect), while the multi-phase learning strategy is a necessary mechanism that allows the mask-free editing-based video dubber to function reliably within this new paradigm.

### F.2. Validation of the Bootstrapping Effect

To further validate the effectiveness of our bootstrapping paradigm, we evaluate the final mask-free model ($\mathcal{G}_{free}$) against the synthetic data used for its own training (curated from $\mathcal{G}_{mask}$). Specifically, we sampled 20 held-out pairs from the constructed dataset and compared them with the outputs of $\mathcal{G}_{free}$ given the same real source videos.

As shown in Tab. 12, $\mathcal{G}_{free}$ consistently outperforms the constructed training data in terms of lip synchronization (Sync-C) and identity preservation (CSIM), while maintaining comparable visual quality (FID). We attribute this improvement to our denoising training objective: by taking potentially flawed synthetic data as input but strictly targeting real video as supervision, the model effectively learns to filter out synthetic artifacts. Consequently, $\mathcal{G}_{free}$ treats the imperfections in the constructed data as noise to be suppressed, resulting in a model that is more robust and produces higher fidelity results than the data it was trained on.

*Table 12.* Quantitative comparison between the constructed training data (generated by $\mathcal{G}_{mask}$) and the output of our final model $\mathcal{G}_{free}$. The "student" ($\mathcal{G}_{free}$) surpasses the "teacher" (data).

| Method | FID $\downarrow$ | Sync-C $\uparrow$ | CSIM $\uparrow$ |
|---|---|---|---|
| Constructed Data (via $\mathcal{G}_{mask}$) | 7.00 | 7.88 | 0.905 |
| **Ours ($\mathcal{G}_{\mathbf{free}}$)** | **6.98** | **8.97** | **0.912** |

### F.3. Ablation on Stage-I Data Generator Quality

To further analyze the role of Stage I in the proposed bootstrapping paradigm, we replace our tailored mask-based data generator $\mathcal{G}_{mask}$ with a weaker generator, LatentSync (Li et al., 2024), and use the resulting pseudo-paired data to train the same Stage-II mask-free video dubber. This controlled experiment isolates the effect of Stage-I data quality while keeping

the Stage-II architecture and training strategy unchanged. As shown in Tab. 13, the results confirm the importance of both stages.

*Table 13.* **Ablation on Stage-I data generator quality.** Metrics are reported as Stage I → Stage II, where Stage II uses the same mask-free dubber trained on pseudo-paired data produced by the specified Stage-I generator.

| Dataset | Stage-I Generator | FID ↓ | LPIPS ↓ | Sync-C ↑ |
|---|---|---|---|---|
| HDTF | LatentSync | 8.04 → 7.22 | 0.024 → 0.019 | 8.16 → 8.50 |
| HDTF | **Ours w/ $\mathcal{G}_{\text{mask}}$** | **7.87 → 7.03** | **0.018 → 0.014** | **8.05 → 8.56** |
| X-DubBench | LatentSync | 13.60 → 10.04 | 0.031 → 0.026 | 6.28 → 7.05 |
| X-DubBench | **Ours w/ $\mathcal{G}_{\text{mask}}$** | **10.82 → 9.35** | **0.026 → 0.023** | **6.51 → 7.28** |

First, a stronger Stage-I generator leads to a better Stage-II dubber. Compared with using LatentSync-generated pairs, our tailored $\mathcal{G}_{\text{mask}}$ produces higher-quality pseudo-paired data and consequently yields stronger final performance on both HDTF and X-DubBench. This validates Stage I as an indispensable enabler that lays the foundation for mask-free training, and also supports the necessity of our dedicated data construction and curation designs.

Second, even when trained on lower-quality pseudo-pairs generated by LatentSync, the Stage-II mask-free dubber still consistently outperforms its Stage-I generator. This demonstrates the effectiveness of the inpainting-to-editing paradigm: by taking potentially imperfect synthetic videos as inputs while using real videos as supervision targets, the Stage-II model learns to suppress synthetic artifacts and leverage complete spatiotemporal context during inference. Therefore, the gains of our final model come from both the quality of the Stage-I data generator and the paradigm shift from mask-based inpainting to mask-free editing.

## G. Additional Analysis on Lip-Shape Leakage and Expression Bias

### G.1. Quantitative Evaluation of Lip-Shape Leakage

Lip-shape leakage is a non-trivial challenge in visual dubbing, where a model may inherit lip-shape cues from the source video rather than generating speech-consistent lip motion from the target audio. In our framework, we mitigate this issue at multiple stages: the Stage-I mask design reduces leakage from mask-boundary variations, the lip-shape distinction filter removes pseudo-paired samples with insufficient lip variation, and the Stage-II SyncNet supervision directly encourages audio-visual synchronization.

Following KeySync (Bigata et al., 2025), we compute LipLeak scores to quantitatively evaluate lip-shape leakage. As shown in Tab. 14, our full model achieves the lowest LipLeak score among baselines and ablations. Compared with the variants without lip-shape filtering or SyncNet supervision, the full model further reduces leakage, validating the effectiveness of the proposed mitigation strategies. This quantitative result is also consistent with the qualitative comparison in Fig. 4, where mask-based methods exhibit open-mouth artifacts during silent frames, while our mask-free dubber produces more speech-consistent lip motion.

*Table 14.* **Quantitative evaluation of lip-shape leakage.** Lower LipLeak scores indicate less leakage from source lip shapes.

| Method | Wav2Lip | VideoReTalking | LatentSync | w/o Filtering | w/o SyncNet | Ours-$\mathcal{G}_{\text{free}}$ |
|---|---|---|---|---|---|---|
| LipLeak ↓ | 0.54 | 0.29 | 0.33 | 0.23 | 0.19 | **0.17** |

### G.2. Source-Video Influence and Residual Expression Bias

The above evidence suggests that the observed source-video influence in some challenging cases is better characterized as residual expression bias rather than persistent phoneme-level lip-shape leakage. Specifically, the lip-sync metrics in Tabs. 1 and 2, together with the LipLeak results in Tab. 14, indicate that phoneme-level lip synchronization is already substantially improved. However, when the source video contains a strong global facial expression, such as a wide smile, the generated result may still inherit part of this overall expression even when the key phoneme-related lip motion is synchronized. A representative example is shown in Fig. 11. When the source video contains a pronounced wide smile, the default audio-driven output still achieves speech-relevant lip motion, but partially preserves the source video's global

smiling expression. This behavior suggests that the remaining source-video influence is more related to residual expression bias than to direct phoneme-level lip-shape leakage.

This observation points to an important future direction for visual dubbing: beyond phoneme-level lip synchronization, modeling controllable or speech-matched global facial expressions remains worthy of further attention. Such expression-level control is complementary to lip synchronization and may require additional conditioning signals beyond pure audio, such as text prompts, emotion labels, or audio-emotion representations.

### G.3. Text-Controlled Expression Exploration

To further examine whether the residual expression bias is inherent to our framework, we conduct a preliminary exploration by introducing an explicit text prompt for expression control during inference, supplementing the standard pure audio-driven dubbing setting. As shown in Fig. 11, we compare three cases: the source video with a pronounced wide smile, our default output using an empty prompt, and our output with an explicit neutral-expression prompt. Although the default output achieves speech-relevant lip motion, it still partially inherits the source video's wide-smile expression. By using the prompt "A woman is singing with a calm and neutral expression," the generated result suppresses the wide-smile influence to some extent and produces a more neutral expression.

This result suggests that residual source-video expression bias is not unavoidable, but can be mitigated by explicit conditioning signals. In the standard audio-driven visual dubbing setup, visual cues from the source video typically dominate audio signals regarding global expressions. By contrast, text-based expression control provides an additional semantic signal that can guide the model toward the desired overall facial expression.

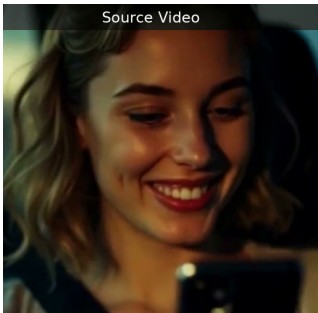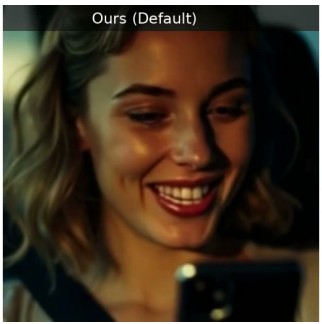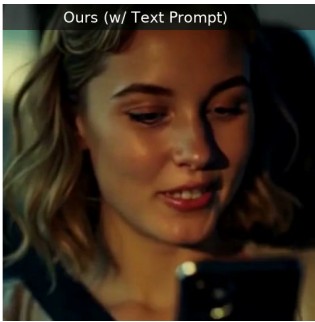

*Figure 11.* **Text-controlled expression exploration.** Left: the source video contains a pronounced wide smile. Middle: our default setting with an empty prompt achieves speech-relevant lip motion but still partially inherits the source smile. Right: adding the text prompt "A woman is singing with a calm and neutral expression" mitigates the residual expression bias and yields a more neutral expression.

### G.4. Scope and Future Directions

For fair comparison with existing visual dubbing methods, all main experiments follow the standard audio-driven setting and use an empty positive prompt during inference, as described in Appendix A.2. Therefore, the results reported in the main paper do not rely on additional text-based expression control. The preliminary exploration above is intended only to analyze the source of residual expression bias and to highlight a promising direction for future work.

In future work, we plan to incorporate more explicit expression control signals into both data construction and the final dubbing model, such as text-controlled expression variation in pseudo-paired data and audio-emotion modeling in Stage II. Overall, these analyses clarify that our generative bootstrapping and mask-free paradigm reduces lip-shape leakage while achieving strong improvements in lip sync, visual quality, identity preservation, and robustness. At the same time, speech-matched global expression control remains an important and promising direction for the broader visual dubbing community.

## H. Calculation of Success Rate

In our quantitative evaluation on X-DubBench (Tab. 2), we report the Success Rate metric. To quantify this, we enlisted 10 participants to review the generated videos and provide a binary judgment (*Success* or *Failure*). Evaluators were instructed to classify a generation as a 'Failure' if it exhibited severe visual collapse or complete lip-synchronization mismatch. Each

video received at least 5 independent evaluations, and the final classification as a successful case was determined by a majority vote.

# I. Details of User Study

The user study involved 30 participants. Each participant received compensation of approximately 15 USD for completing a session that lasted 40–50 minutes, which aligns with the average hourly wage. For reference, Fig. 12 provides screenshots of the rating interface used in the study.

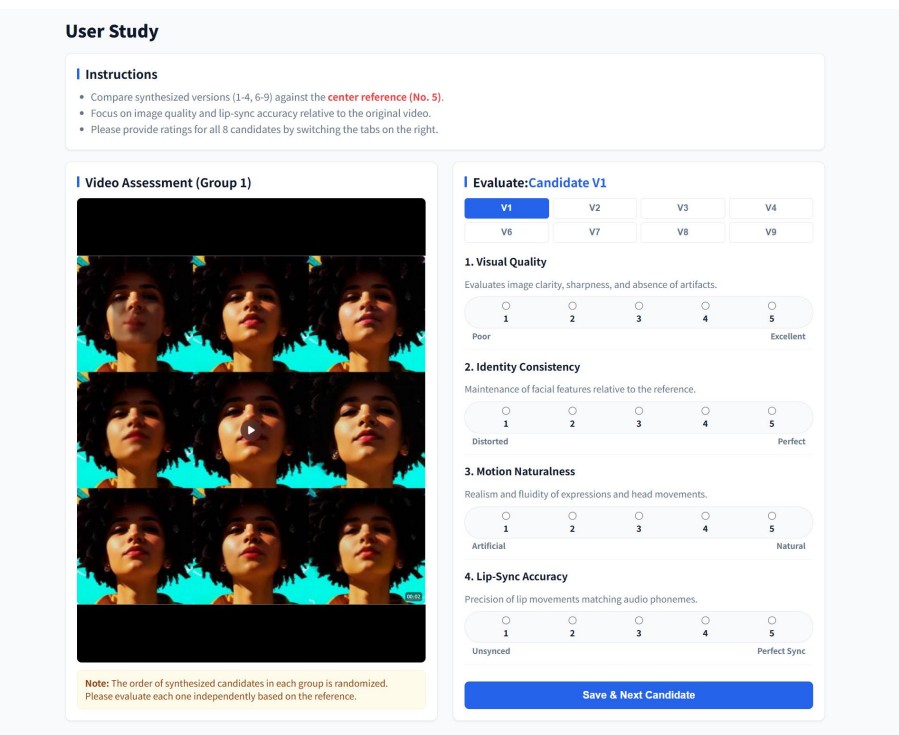

*Figure 12.* Screenshot of the rating interface of the user study.

# J. Details of X-DubBench

To thoroughly evaluate our framework, we construct X-DubBench benchmark, a challenging benchmark comprising 440 video-audio pairs. The dataset is carefully designed with the following composition:

**Audio data.** The audio component includes both speech and singing. For speech, we randomly sampled 350 clips from Common Voice (Ardila et al., 2019), spanning six languages and dialects: 170 in English, 60 in Mandarin, 30 in Cantonese, 30 in Japanese, 30 in Russian, and 30 in French. For singing, we incorporated 60 English clips from NUS-48E (Duan et al., 2013) and 30 Mandarin clips from Opencpop (Wang et al., 2022). Each segment lasts between 7 and 14 seconds and captures a wide range of speaking rates, pitch levels, accents, and vocal styles, ensuring rich phonetic and linguistic diversity.

**Video data.** The video set combines real-world recordings and AI-generated content from publicly available sources with proper copyright clearance (e.g., Civitai, Mixkit, Pexels). It contains 291 clips of natural human subjects, 108 clips of stylized characters with distinct artistic features, and 41 clips of non-human or humanoid entities with durations ranging from 2 to 9 seconds. Representative samples are shown in Fig. 13, Fig. 14, and Fig. 15. Unlike conventional datasets, which are typically captured under controlled conditions, X-DubBench is explicitly designed to reflect real-world challenges. The dataset incorporates dynamic lighting, partial occlusions, identity-preserving transformations, and substantial variations in pose and motion. By embedding these factors, X-DubBench more faithfully captures the diversity and unpredictability of real-world scenarios, providing a rigorous testbed for evaluating lip-synchronization models. Illustrative examples are shown in Fig. 16.

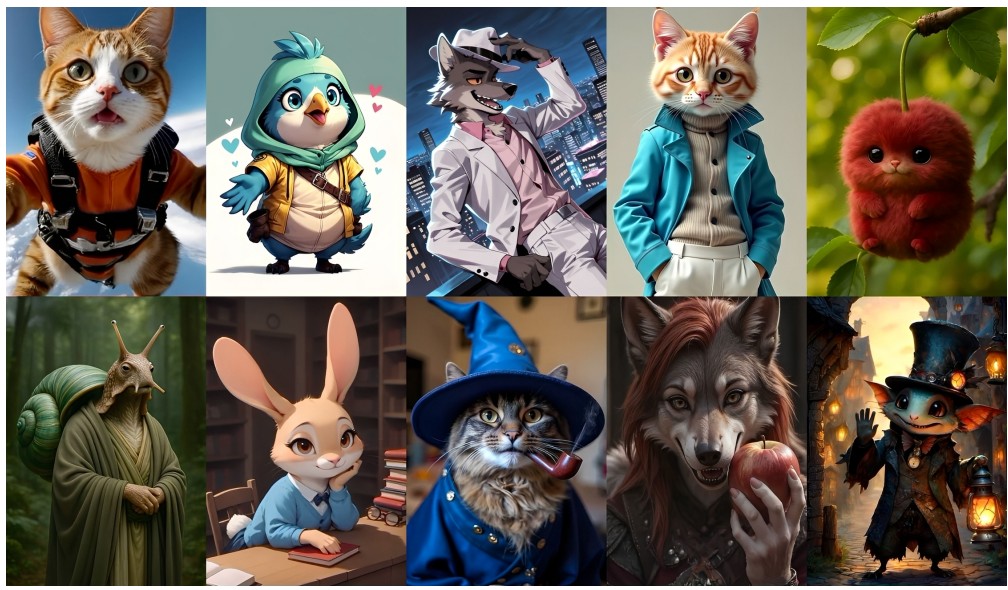

*Figure 13.* X-DubBench benchmark examples (I): non-human characters with diverse morphological variations.

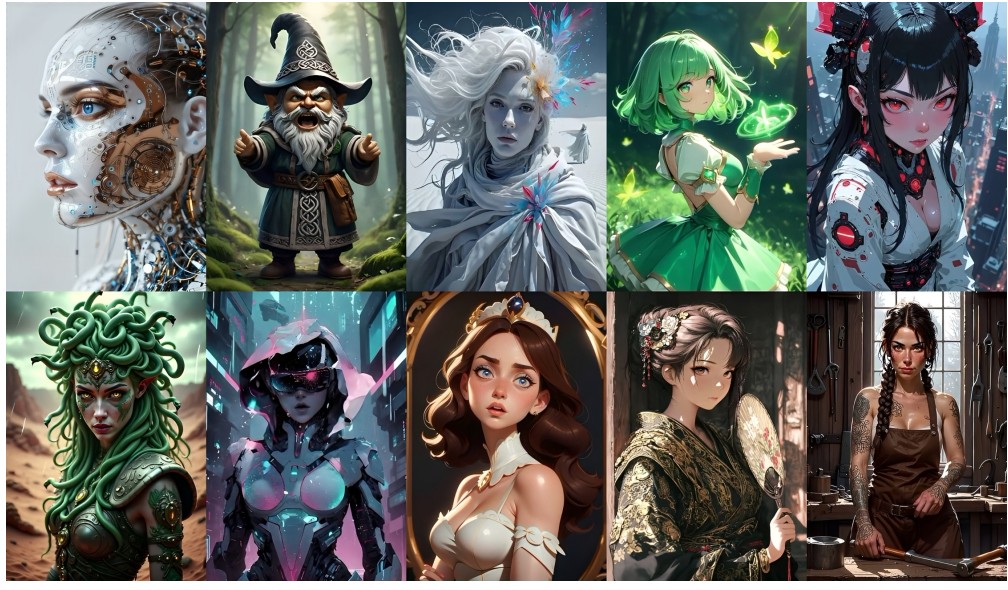

*Figure 14.* X-DubBench benchmark examples (II): stylized characters with distinctive visual designs.

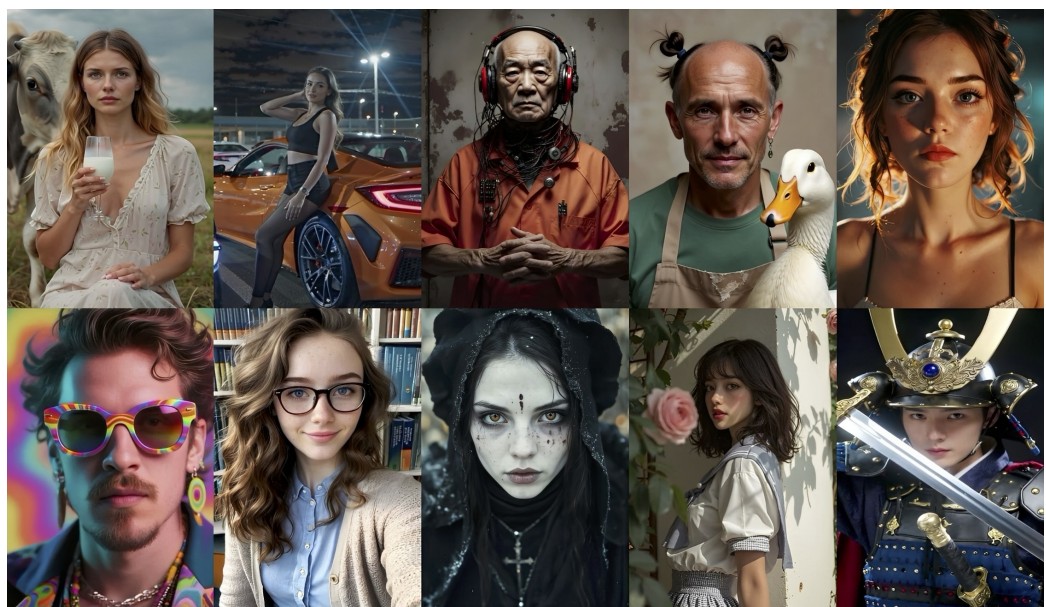

*Figure 15.* X-DubBench benchmark examples (III): real-world human appearances in practical conditions.

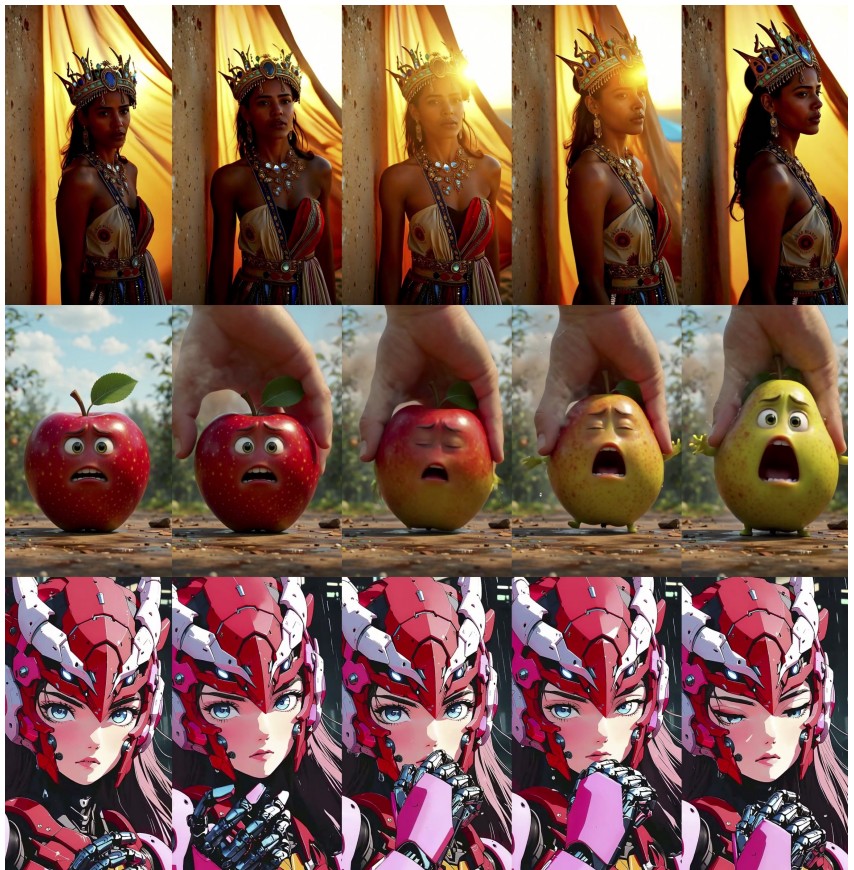

*Figure 16.* Additional samples from X-DubBench showing lighting variations, identity-preserving changes, and occlusions in complex real-world scenarios.

