# OpenReview forum: "From Inpainting to Editing: Unlocking Robust Mask-Free Visual Dubbing via Generative Bootstrapping"
_ICML.cc/2026/Conference — ICML 2026 regular_

### Official Review · Reviewer_5e9i · 2026-03-11

**Soundness:** 3
**Presentation:** 3
**Significance:** 3
**Originality:** 3
**Overall Recommendation:** 5
**Confidence:** 5

**Summary:**

This paper proposes X-Dub, a two-stage generative bootstrapping framework for mask-free visual dubbing. The method repurposes a mask-based inpainting model to generate pseudo-paired data, which is then used to train a mask-free editing model. The authors also introduce timestep-adaptive multi-phase learning and X-DubBench benchmark.

**Compliance With Llm Reviewing Policy:**

Affirmed.

**Final Justification:**

This paper proposes a two-stage training paradigm for mask-free video dubbing. The authors have constructed an exceptionally sophisticated data generation pipeline, enabling the acquisition of large-scale, high-quality paired training data. The visual results demonstrated in the experiments are highly compelling. Regarding the concern I raised during the rebuttal phase about expression leakage from reference videos, the authors have proposed a highly effective solution.
In summary, my final recommendation is Accept.

**Key Questions For Authors:**

1. How do you address the lip shape leakage problem where the generated results are biased toward the original video's mouth shapes? Have you considered any debiasing strategies?
2. What specific technical innovations distinguish this two-stage framework from simple cascaded training?
3. Using editing methods to modify lip movements doubles the sequence length involved in attention computation. However, since most of the information provided by the original video is already temporally aligned, this approach leads to increased inference costs. Have you considered other more efficient alternatives?

**Limitations:**

yes

**Strengths And Weaknesses:**

Strengths
1. Meticulous data processing pipeline: Creative use of VL models for occlusion detection and 3D rendered data to address non-homologous issues.
2. Strong visual quality: The method achieves good qualitative results across diverse scenarios.
3. Benchmark contribution: X-DubBench provides a valuable evaluation resource for the community.
Weaknesses
1. Limited novelty: The two-stage training paradigm lacks significant academic innovation.
2. Lip shape leakage issue: The pseudo-paired data generated by stage-1 inherits lip shapes from the original video, which propagates to stage-2. For example: In supplementary video 3:33, the girl maintains a wide smile because the source video shows laughing. At 3:40, the anime character barely opens the mouth as the original keeps lips closed.

---

> ### Author Rebuttal · Authors · 2026-03-31
>
> We appreciate your positive and detailed feedback, and address concerns below.
>
> ---
> > **Q1 & W5: Strategies to Mitigate Lip-Shape Leakage**
>
> Thanks for this important question. Lip-shape leakage is indeed a non-trivial challenge in visual dubbing. To tackle this, we implemented dedicated strategies throughout the framework.
> - **Stage I Mask Design:** As detailed in Appendix A.3, instead of commonly used masks that may leak lip-shape cues through boundary changes or constrain jaw movement, we use carefully designed 3DMM-derived masks with a fixed jaw-opening parameter and morphological dilation, reducing Stage I leakage.
> - **Lip-Shape Distinction Filtering:**
> As described in line 245 and Appendix B.6, we filter pseudo-paired samples using normalized mouth landmark distance and mouth aspect ratio (vertical lip opening / mouth-corner width). This excludes pairs with insufficient lip-shape differences and reduces leakage propagation to Stage II.
> - **Stage II SyncNet Supervision:** In Stage II, we incorporate a SyncNet loss to directly enforce sync between lip motion and target speech.
> This encourages speech-consistent lip motion rather than inheriting lip-shape or expression bias from the input video.
>
> **Quantitative Evaluation:**
> Following KeySync [Bigata et al., 2025], we compute LipLeak scores to quantitatively evaluate lip-shape leakage (against baselines & ablations of the above strategies):
>
> |Method|Wav2Lip|VideoReTalking|LatentSync|w/o Filtering|w/o SyncNet|Full Ours ($\mathcal{G}_{free}$)|
> |:-|:-:|:-:|:-:|:-:|:-:|:-:|
> |**LipLeak ↓**|0.54|0.29|0.33|0.23|0.19|**0.17**|
>
> The above strategies reduce LipLeak, and our final model achieves the lowest score, consistent with cases in Fig. 4 (rightmost column). We further attribute this to our Stage II cross-pair supervised training with real synchronized videos as targets, rather than mask-based self-reconstruction.
>
> **Expression Bias & Future Directions:**
> We fully agree with your insightful observation that, in some cases, the generated results may retain the overall expression bias of the source video (e.g., a wide smile), even when key phoneme-related lip motions are already synchronized. This points to the next step for visual dubbing: going beyond phoneme-level lip sync to model speech-matched expressions, e.g., via pseudo-pairs with text-controlled expression variation and audio-emotion modeling in Stage II.
> We will include this discussion in the revised manuscript.
>
> ---
> > **Q2 & W4: Innovation & Distinction from Cascaded Methods**
>
> We appreciate this point. As emphasized in the paper, our key contribution is not simply the two-stage design itself, but **the formulation shift and learning paradigm** it enables for visual dubbing:
>
> **Reformulation of Visual Dubbing:** We are the first in visual dubbing to reframe the task from restrictive mask-based inpainting to mask-free editing, fundamentally aligned with the nature of dubbing.
>
> **Data and Learning Paradigm:** To make this shift feasible, we propose a novel data construction pipeline that builds pseudo-paired videos from abundant unlabeled in-the-wild audiovisual data, enabling supervised training for mask-free dubbing.
>
> **Distinction from Cascaded Methods:**
> Our final model performs end-to-end mask-free inference with full spatiotemporal context, whereas cascaded methods (e.g., VideoReTalking and IP-LAP) remain mask-dependent and rely on multi-module sequential inference. Thus, we achieve better lip sync, identity consistency, and especially robustness in challenging cases like occlusions.
>
> ---
> > **Q3: Efficient Alternatives**
>
> Yes. Beyond our token-concatenation conditioning with full attention, we considered alternatives: channel concatenation (already ablated in Sec. 4.3; half the attention token length) and a sparse 3D-attention variant, which restricts each generated frame to attend only to its temporally corresponding input frame.
>
> |Method|Complexity|FID ↓|Sync-C ↑|CSIM ↑|
> |:-|:-:|:-:|:-:|:-:|
> |Channel Concat|$O((fhw)^2)$|8.89|7.49|0.873|
> |Sparse 3D Attention|$O(2(fhw)^2+f(hw)^2)$|7.41|8.50|0.878|
> |Token Concat (Ours)|$O((2fhw)^2)$|**7.03**|**8.56**|**0.883**|
>
> - **Channel concatenation** is the most efficient, but significantly degrades lip sync, visual quality, and identity consistency, because conditioning is limited to shallow fusion and enforces rigid spatial alignment.
> - **Sparse 3D attention** offers a better efficiency–quality trade-off, but still degrades visual quality because each generated frame receives more limited input context.
> - **Token concatenation** achieves the best overall quality and is therefore used in our final model.
>
> In practice, the cost of the final setting can be further reduced using the acceleration strategies in Appendix E, while sparse 3D attention provides a higher-efficiency alternative under relaxed quality requirements.
> In addition, we operate on cropped face regions rather than full images, avoiding extra attention cost on large irrelevant backgrounds.

---

> > ### Author Rebuttal · Reviewer_5e9i · 2026-04-02
> >
> > Although the authors have proposed several measures to mitigate lip leakage, they also acknowledge that the current method remains susceptible to the influence of the reference video, which constitutes one of the limitations of this work. Regarding the other concerns, the authors have provided comprehensive supplementary experiments. Therefore, I will maintain my original score.

---

> > > ### Author Response · Authors · 2026-04-07
> > >
> > > We sincerely thank the reviewer for your careful review and continued engagement throughout the discussion. Your close scrutiny helps us better present the scope of our current method and its future directions.
> > >
> > > ---
> > > > **On the Remaining Concern: Source-Video Influence and Residual Expression Bias**
> > >
> > > **Our Reading of the Existing Evidence:** Given the existing quantitative results (the lip-sync metrics in Tabs. 1 & 2 and the LipLeak score provided in our previous reply) and the specific cases you pointed out, where key phoneme-level lip sync is already achieved but the result tends to inherit the source video's overall expression, we believe that the observed "source-video influence" is better characterized as residual expression bias, rather than the persistent phoneme-level lip-shape leakage commonly seen in visual dubbing.
> > >
> > > This also suggests that, beyond phoneme-level lip sync, modeling controllable or speech-matched overall facial expressions remains worthy of further attention in this field.
> > >
> > > **Additional Exploration:**
> > > To mitigate this expression bias, we further explore introducing an explicit text prompt for expression control during inference, supplementing the standard pure audio-driven dubbing setting. A supplementary video result for the wide-smile case is provided here: **[[video link]](https://anonymous.4open.science/api/repo/icml-22928/file/5e9i/5e9i.mp4?v=35edecd4)** (text prompt: _"A woman is singing with a calm and neutral expression."_).
> > >
> > > - **Key Finding:** The explicit text control suppresses the source video's "wide smile" influence to some extent, yielding a more neutral expression in the output video.
> > > - **Interpretation:** In a standard audio-driven visual dubbing setup, visual cues from the source video typically dominate audio signals regarding global expressions. The preliminary result above demonstrates that **the residual source-video expression bias is not unavoidable** in our framework, and can be partially alleviated by explicit conditioning signals.
> > >
> > > **Scope and Next Steps:**
> > > For a fair comparison with existing visual dubbing methods, we did not introduce additional explicit text signals in the main paper (using an empty prompt, as detailed in Appendix A.2). However, the preliminary exploration above, while not yet flawless, provides initial evidence that our method is not inherently constrained by expression bias.
> > > It also points to a promising next step: introducing more explicit expression control signals (e.g., text prompts or audio-emotion cues) during data construction and in the final dubbing model.
> > >
> > > **Final Remarks:**
> > > We hope this discussion helps clarify that, while speech-matched expression control remains an important direction for future work, it **does not diminish our core conclusion within the current visual dubbing setting**: the proposed generative bootstrapping and mask-free paradigm yield stable and significant improvements in lip sync, visual quality, identity preservation, and robustness, while also reducing lip-shape leakage.
> > >
> > > ---
> > > Once again, we sincerely thank you for your insightful perspective and rigorous scrutiny. We will reflect this valuable discussion more clearly in the revised manuscript and continue to refine this direction in our ongoing research.

---

### Official Review · Reviewer_ubN8 · 2026-03-13

**Soundness:** 3
**Presentation:** 3
**Significance:** 2
**Originality:** 3
**Overall Recommendation:** 4
**Confidence:** 3

**Summary:**

This paper proposes a two-stage generative bootstrapping framework named X-Dub , aiming to address issues in audio-driven visual dubbing caused by mask dependency, such as lip-shape leakage, identity drift, and poor robustness to occlusions. Since it is virtually impossible to obtain real-world paired video data differing only in lip motion , the first stage of this method specifically repurposes a mask-guided inpainting model as a dedicated data generator. It utilizes Flow Matching and Diffusion Transformers (DiTs) to synthesize large-scale, high-fidelity pseudo-paired data. In the second stage, the authors use these pseudo-paired data to train a mask-free video editing model as the final dubber , enabling it to directly process complete video inputs during inference and thereby break free from masking artifacts. Furthermore, the paper introduces a timestep-adaptive multi-phase learning strategy , which disentangles the optimization objectives of structure, lip motion, and texture across different diffusion phases , and constructs a new benchmark dataset called X-DubBench that includes complex real-world scenarios.

**Compliance With Llm Reviewing Policy:**

Affirmed.

**Final Justification:**

I keep my score at "Weakly Accept".

**Key Questions For Authors:**

1. What are the specific settings for $\lambda_{face}$ and $\lambda_{lip}$ in Equation (1)? How sensitive is the model's performance to these spatial weight factors?

2. Please clarify whether the baseline models (such as LatentSync, Diff2Lip, etc.) in Table 1 and Table 2 directly utilized their official pre-trained weights, or were they retrained on the same 600-hour internet dataset as your method?

3. Considering practical applications, if the acceleration techniques mentioned in Appendix E (reducing the number of denoising steps to 25 and incorporating TeaCache) are utilized to compress the inference time to 25 seconds, what specific degradation would occur in the model's core metrics (e.g., Sync-C, FID, LPIPS) on the HDTF and X-DubBench datasets? Please provide quantitative results.

**Limitations:**

Yes

**Strengths And Weaknesses:**

Strengths

1. The generative bootstrapping paradigm presented in this paper demonstrates notable innovation. The idea of downgrading the mask-based model from the "primary inference engine" to a "training data generator," thereby bootstrapping a mask-free editing model, cleverly breaks the bottleneck of paired data dependency in the visual dubbing domain. Simultaneously, the timestep-adaptive multi-phase learning strategy (which divides the denoising process into high-noise structure learning, mid-noise lip editing, and low-noise texture fine-tuning) theoretically aligns well with the frequency bias characteristics of diffusion models, and its engineering implementation—via attaching different LoRA experts—is also quite elegant.

2. The experimental evaluation is detailed and comprehensive. The authors not only validated their method on the standard HDTF dataset but also constructed the highly challenging X-DubBench benchmark, which covers complex scenarios such as dynamic lighting, occlusions, and stylized characters. The ablation studies are rigorously designed; in particular, the decoupled ablation regarding the paradigm (inpainting vs. editing) and the training strategy (single-phase vs. multi-phase) in Table 10 clearly proves the independent contributions of each module.

3. The paper features a rigorous structure and clear motivation. It transitions naturally from the physical destructiveness of existing mask-based methods (such as the destruction of spatiotemporal context) to the data scarcity pain point inherent in mask-free V2V learning, forming a highly cohesive and sound logical loop.

Weaknesses.

1.The model's final performance remains limited, to some extent, by the upper bound of the quality of the pseudo-paired data generated in the first stage.

2.In the single-step denoising truncation strategy (Eq. 12), the threshold is empirically set to 0.6, lacking sufficient foundational theoretical support.

3.According to Table 9, the model (1.5B parameters) requires approximately 60 seconds (1 minute) to process a 3-second video on an A100 GPU, indicating an excessively high computational cost compared to Wav2Lip (1s) or LatentSync (30s). In the practical deployment of visual dubbing, which highly depends on real-time or near-real-time interaction, there is a lack of detailed evaluation regarding the quality degradation after acceleration.

---

> ### Author Rebuttal · Authors · 2026-03-30
>
> We sincerely thank the reviewer for your detailed and constructive feedback, as well as the positive assessment of our method's innovation and motivation. We address the remaining concerns below.
>
> ---
> > **W1: Relationship Between Final Performance and Stage I Data Quality**
>
> We appreciate your insightful observation on data dependency. We agree that Stage I data quality is crucial and positively correlates with Stage II performance. However, empirical evidence shows that **our final model is not strictly upper-bounded by Stage I data quality**:
>
> * **Stage II vs. Stage I (Tabs. 1 & 2):** Stage II dubber ($\mathcal{G}_{free}$) consistently outperforms the generic Stage I data generator ($ \mathcal{G}\_{mask}^*$) (e.g., ΔFID = -0.84/-1.47, ΔSync-C = +0.52/+0.77).
> * **Final Model vs. Curated Data (Appendix F.2, Tab. 11):** Final $\mathcal{G}_{free}$ even surpasses the carefully curated Stage I generated data (post-quality filtering) on samples unseen during Stage II training (e.g., ΔFID = -0.02, ΔSync-C = +1.09).
>
> This validates our "bootstrapping" paradigm: Stage II inherently learns to suppress synthetic artifacts by anchoring on pristine real videos as supervision targets. More fundamentally, the shift from mask-based inpainting to mask-free editing allows the final model to leverage full spatiotemporal context, ultimately yielding superior dubbing performance.
>
> We acknowledge that, like all data-driven methods, our final performance is inevitably influenced by the data quality. Nevertheless, our current paradigm mitigates this dependency to some extent and achieves strong empirical gains. Future work will explore post-training strategies to further push beyond the limitations of static supervised training.
>
> ---
> > **W2: Justification for the Denoising Truncation Threshold**
>
> Thank you for this important question. The threshold of 0.6 in Eq. 12 was carefully determined through early experiments:
>
> * **Qualitative:** **[Single-step denoising results](https://anonymous.4open.science/r/icml-22928/ubN8/ubN8-W2.mp4)** below ~0.6 remain clear and reliable enough for pixel-level supervision, whereas those at higher noise levels become increasingly blurry and unstable.
>
> - **Quantitative:** We computed the MSE between clean frames and single-step denoising reconstructions across noise levels. As shown in this **[comparative plot](https://anonymous.4open.science/r/icml-22928/ubN8/ubN8-W2.png)**, the no-truncation MSE increases noticeably beyond ~0.6, indicating less reliable predictions; after applying the 0.6 threshold, the MSE remains bounded within the low-error region, preventing noisy gradients from destabilizing training.
>
> We will include the above analysis in the revised manuscript.
>
> ---
> > **W3 & Q3: Quantitative Evaluation of Inference Acceleration**
>
> We appreciate this practical consideration. Below we report quantitative comparisons before and after acceleration (25 denoising steps + TeaCache), showing negligible degradation (***Format:** Original → Accelerated*):
>
> |Dataset|FID ↓|LPIPS ↓|Sync-C ↑|
> |:-|:-:|:-:|:-:|
> |HDTF|7.03 → 7.18|0.014 → 0.016|8.56 → 8.56|
> |X-DubBench|9.35 → 9.41|0.023 → 0.029|7.28 → 7.25|
>
> **Key Findings:** After acceleration, Lip sync (Sync-C) remains practically unaffected; visual quality (FID, LPIPS) shows only marginal drops and still clearly outperforms baselines in Tabs. 1 & 2.
> **Our acceleration strategies improve practical deployment efficiency with only minimal quality degradation**.
>
> ---
> > **Q1: Loss Weight Settings and Sensitivity**
>
> We empirically set $\lambda_{face}=\lambda_{lip}=0.6$ to balance the three loss terms.
>
> Early experiments showed that the model is relatively insensitive to loss weights within roughly 0.4~1.0, while more extreme values degraded lip sync (if too small) or visual quality (if too large).
>
> We will clarify this empirical choice in the revised manuscript.
>
> ---
> > **Q2: Baseline Configurations and Retrained LatentSync**
>
> We evaluated baselines using their official pretrained weights, consistent with the setup adopted in most prior works. We will explicitly state this setup in the revised manuscript.
>
> We thank the reviewer for this valuable suggestion and retrained the strongest baseline, LatentSync, on the same 600-hour internet dataset. As shown below, retraining brings only limited changes and still leaves a clear gap to our method (***Format:** Official → Retrained*).
>
> |Dataset|Method|FID ↓|LPIPS ↓|Sync-C ↑|
> |:-|:-|:-:|:-:|:-:|
> |HDTF|LatentSync|8.04 → 7.97|0.024 → 0.026|8.16 → 8.00|
> ||Ours ($\mathcal{G}_{free}$)|**7.03**|**0.014**|**8.56**|
> |X-DubBench|LatentSync|13.60 → 12.83|0.031 → 0.028|6.28 → 6.43|
> ||Ours ($\mathcal{G}_{free}$)|**9.35**|**0.023**|**7.28**|
>
> The complete retrained LatentSync results are available **[here](https://anonymous.4open.science/r/icml-22928/ubN8/ubN8-Q2.png)** and will be incorporated into the main comparison table.
>
> ---
> Thanks again for your valuable suggestions, which greatly help strengthen our work.

---

> > ### Author Rebuttal · Reviewer_ubN8 · 2026-04-02
> >
> > I have no more concerns, and maintain my positive rating.

---

> > > ### Author Response · Authors · 2026-04-05
> > >
> > > Thank you for your appreciation of our work and your positive response to our rebuttal. We sincerely appreciate your careful review and effort throughout the discussion. Your feedback helps us a lot in refining our final manuscript.

---

### Official Review · Reviewer_VtsZ · 2026-03-14

**Soundness:** 4
**Presentation:** 4
**Significance:** 3
**Originality:** 3
**Overall Recommendation:** 5
**Confidence:** 4

**Summary:**

This paper studies the problem of visual dubbing. Visual dubbing refers to synchronizing a video's lip movements with new speech. The task is challenging because it requires training data in which the same person appears while only the lip motion changes. Observing that existing methods often rely on mask-based inpainting to address this problem and may ignore the spatial-temporal context of the masked region, the paper proposes a two-stage procedure. In the first stage, a mask-guided inpainting model is trained on large-scale data. In the second stage, the first-stage model is used as a generator to produce synthetic-real video data pairs in which the lip movements differ. Experiments demonstrate that this method achieves strong performance compared with existing baselines.

**Compliance With Llm Reviewing Policy:**

Affirmed.

**Final Justification:**

I believe the rebuttal addresses my concerns regarding qualitative comparisons and ablations. The authors also provides clarification regarding the efficiency issues. Given the clear qualitative improvements, I would like to maintain my positive ratings and increase my confidence to 4.

**Key Questions For Authors:**

1. (For Weakness 1) Would it be possible to include video-level qualitative comparisons with baseline methods on the demo page? (If the rebuttal policy does not allow updating the web page in suppl, I will not weigh this weakness heavily in the final evaluation.)
2. (For weakness 2) How important is Stage 1 for creating the training data? Would it be possible to include an ablation study where Stage 1 is removed or replaced with a weaker Stage 1 generator?

**Limitations:**

Yes.

**Strengths And Weaknesses:**

Strengths
1. The paper proposes a clear and well-motivated methodology: first leveraging an inpainting model to generate synthetic paired data, and then training a mask-free visual dubbing model using this data.
2. The paper makes a solid effort to reduce potential bias introduced by the first-stage generator. It employs in-domain creation, performs quality filtering, and applies data augmentation to improve robustness under diverse scenarios.
3. The paper presents results for both stages of the framework, clearly demonstrating the contribution of each stage.
4. The demo videos provided in the supplementary material are impressive. The results appear robust across different languages, noisy video inputs, and long videos.



Weakness
1. Regarding result presentation, although the paper provides qualitative comparisons in the manuscript, it would be helpful if the demo page could include video-level qualitative comparisons with baseline methods.
2. While the paper includes a number of ablation studies, it does not directly analyze the importance of introducing Stage 1. The Stage 1 model itself already achieves strong performance (often ranking second and only slightly weaker than the final model). It would be useful to evaluate the performance when Stage 1 is removed or replaced with a weaker generator.
3. Efficiency: In Table 9, the efficiency result shows the model cannot perform a real-time visual dubbing.
4. (Minor) It is worth noting that the paper’s contribution more lies in training data preparation rather than new technical module/algorithm innovation. However, I would like to comment that this is not a strong weakness to me, since the data paradigm appears to be more and more important when training large models in various domains.

---

> ### Author Rebuttal · Authors · 2026-03-30
>
> We sincerely thank the reviewer for your thorough evaluation and constructive feedback. We address each concern point by point below.
>
> ---
> > **W1 & Q1: Video Comparisons**
>
> Thank you for your positive feedback on our demo video. Video comparisons against baselines were also provided in our [original supplementary materials](https://openreview.net/attachment?id=PfhSH42cCy&name=supplementary_material): side-by-side results on X-DubBench (demo video 2:20-3:28) and on HDTF (`index.html`, below the demo video).
>
> For your convenience, we also provide video comparisons **[here](https://anonymous.4open.science/r/icml-22928/VtsZ/demo_comparison.mp4)** for easier access and to further highlight the superior performance of X-Dub over the baselines.
>
> ---
> > **W2 & Q2: Ablations on Stage I**
>
> Thank you for this valuable suggestion. As suggested, we ablate Stage I by replacing our tailored generator $\mathcal{G}_{mask}$ with a weaker one (LatentSync). Results confirm the importance of both stages (***Format:** Stage I → Stage II*):
> |Dataset|Stage I Setting|FID ↓|LPIPS ↓|Sync-C ↑|
> |:-|:-|:-:|:-:|:-:|
> |HDTF|w/ LatentSync|8.04 → 7.22|0.024 → 0.019|8.16 → 8.50|
> ||**w/ $\mathcal{G}_{mask}$ (Ours)**|7.87 → 7.03|0.018 → 0.014|8.05 → 8.56|
> |X-DubBench|w/ LatentSync|13.60 → 10.04|0.031 → 0.026|6.28 → 7.05|
> ||**w/ $\mathcal{G}_{mask}$ (Ours)**|10.82 → 9.35|0.026 → 0.023|6.51 → 7.28|
>
> **Key Findings:**
> * **Importance of Stage I**: A stronger Stage I generator (our tailored $\mathcal{G}_{mask}$) yields a better Stage II dubbing model, validating Stage I as an indispensable enabler that lays the foundation for Stage II mask-free training. This underscores the necessity of our dedicated Stage I design and extensive data construction efforts.
> * **Superiority of Stage II (Paradigm Shift)**: Even with lower-quality data from a weaker generator (LatentSync), Stage II still consistently outperforms Stage I, showing even more pronounced gains than in our final setting. This proves Stage II inherently learns to suppress synthetic artifacts by anchoring real videos as supervision targets. More fundamentally, the gains stem from the "inpainting to editing" transition, which liberates the final model from masking constraints and benefits from leveraging complete spatiotemporal context during inference.
>
> ---
> > **W3: Inference Efficiency and Real-Time Considerations**
>
> We appreciate your practical concern regarding efficiency. It's true that real-time performance remains a general challenge for DiT-based methods due to iterative denoising and attention computation.
>
> **Validated Acceleration Strategies:** As detailed in Appendix E (lines 1037-1044), we can safely reduce the denoising steps (50→25) because the complete input video provides rich and mostly aligned context for the final dubbing model. By further incorporating training-free acceleration techniques (e.g., TeaCache), we achieved a **~25-second inference time for a 3-second clip** without noticeable quality degradation, substantially mitigating practical deployment limitations.
>
> **Future Directions for Real-Time Inference:** Recent advances in few-step distillation (e.g., DMD [1]) and autoregressive paradigms with KV-cache (e.g., Self Forcing [2]) suggest promising directions toward real-time streaming generation, even for much larger DiT models (14B). Integrating these techniques into our framework is highly feasible and forms part of our ongoing research.
>
> [1] Yin et al. Distribution Matching Distillation. CVPR 2024.
>
> [2] Huang et al. Self Forcing. NeurIPS 2025.
>
> ---
> > **W4: Methodological Innovation and Data Paradigm**
>
> Thank you for recognizing our data preparation contribution, which indeed forms a core part of our methodological innovation: **a new formulation and learning paradigm for visual dubbing**.
>
> **Formulation Shift for Visual Dubbing:** We are the first in visual dubbing to reframe the task from restrictive mask-based inpainting to mask-free editing, which is more naturally aligned with the essence of dubbing itself.
>
> **Data Construction Enabling a New Learning Paradigm:** To make this reformulation practical, we introduce a novel data construction pipeline that builds pseudo-paired videos from abundant unlabeled in-the-wild audiovisual resources, thereby enabling supervised training for mask-free dubbing.
>
> **Tailored Algorithmic Design:** To fully unleash the potential of this data paradigm, we propose timestep-adaptive multi-phase learning to disentangle structure, lip motion, and texture across diffusion phases, further strengthening the proposed mask-free editing framework.
>
> We also agree with your insightful point that **data paradigms are increasingly important**, and believe our generative data construction paradigm can provide a viable solution for broader generation tasks (e.g., video object removal).
>
> ---
> Thanks again for your time and effort in reviewing our work. We will incorporate the above clarifications and analyses into the revised manuscript.

---

> > ### Author Rebuttal · Reviewer_VtsZ · 2026-04-02
> >
> > Thank you for the rebuttal and reference to the qualitative comparisons between baselines. my concerns are resolved, and I maintain my positive ratings.

---

> > > ### Author Response · Authors · 2026-04-05
> > >
> > > Thank you sincerely for your supportive evaluation and thoughtful engagement throughout the discussion. Thanks again for the time and effort you devoted to this review. Your feedback greatly helps us strengthen our final manuscript.

---

### Decision · Program_Chairs · 2026-04-30

**Decision:**

Accept (regular)

**Comment:**

This paper introduces a framework that advances audio-driven visual dubbing by shifting the paradigm from mask-based inpainting to mask-free video editing. To overcome the fundamental lack of paired training data, the authors utilize a generative bootstrapping approach. They first employ a mask-guided DIT to generate large-scale pseudo-paired data. This data is subsequently used to train a mask-free dubbing model that processes the complete video frame. The work also introduces a timestep-adaptive multi-phase training strategy and a new evaluation benchmark.

The reviewers reached a unanimous consensus (5,4,5). Reviewer's highlighted the effectiveness of repurposing a mask-based model as a data generator to break the paired-data bottleneck. They appreciated extensive experimental evaluations as well.

AC agrees that the paper presents a technically well-motivated contribution and recommends acceptance.